# Smart Drug Delivery Systems Based on Cyclodextrins and Chitosan for Cancer Therapy

**DOI:** 10.3390/ph18040564

**Published:** 2025-04-13

**Authors:** Larisa Păduraru, Alina-Diana Panainte, Cătălina-Anișoara Peptu, Mihai Apostu, Mădălina Vieriu, Tudor Bibire, Alexandru Sava, Nela Bibire

**Affiliations:** 1Department of Analytical Chemistry, Faculty of Pharmacy, “Grigore T. Popa” University of Medicine and Pharmacy from Iasi, 16th Universitatii Street, 700116 Iasi, Romania; larisa-paduraru@umfiasi.ro (L.P.); mihai.apostu@umfiasi.ro (M.A.); madalina.vieriu@umfiasi.ro (M.V.); alexandru.i.sava@umfiasi.ro (A.S.); nela.bibire@umfiasi.ro (N.B.); 2Department of Natural and Synthetic Polymers, Faculty of Chemical Engineering and Environmental Protection, “Gheorghe Asachi” Technical University of Iasi, 71st Prof. Dr. Docent Dimitrie Mangeron Street, 700050 Iasi, Romania; 3“St. Spiridon” County Clinical Emergency Hospital, 1st Independentei Blvd., 700111 Iasi, Romania; tudor_cd_bibire@d.umfiasi.ro

**Keywords:** cyclodextrins, chitosan, nanoparticles, anticancer, folate receptors, biopolymers, drug delivery, glutathione

## Abstract

Despite improvements in therapeutic approaches like immunotherapy and gene therapy, cancer still remains a serious threat to world health due to its high incidence and mortality rates. Limitations of conventional therapy include suboptimal targeting, multidrug resistance, and systemic toxicity. A major challenge in current oncology therapies is the development of new delivery methods for antineoplastic drugs that act directly on target. One approach involves the complexation of antitumor drugs with cyclodextrins (CDs) and chitosan (CS) as an attempt to counteract their primary limitations: low water solubility and bioavailability, diminished in vitro and in vivo stability, and high dose-dependent toxicity. All those drawbacks may potentially exclude some therapeutic candidates from clinical trials, thus their integration into smart delivery systems or drug-targeting technologies must be implemented. We intended to overview new drug delivery systems based on chitosan or cyclodextrins with regard to the current diagnosis and cancer management. This narrative review encompasses full-length articles published in English between 2019 and 2025 (including online ahead of print versions) in PubMed-indexed journals, focusing on recent research on the encapsulation of diverse antitumor drugs within those nanosystems that exhibit responsiveness to various stimuli such as pH, redox potential, and folate receptor levels, thereby enhancing the release of bioactive compounds at tumor sites. The majority of the cited references focus on the most notable research, studies of novel applications, and scientific advancements in the field of nanostructures and functional materials employed in oncological therapies over the last six years. Certainly, there are additional stimuli with research potential that can facilitate the drug’s release upon activation, such as reactive oxygen species (ROS), various enzymes, ATP level, or hypoxia; however, our review exclusively addresses the aforementioned stimuli presented in a comprehensive manner.

## 1. Introduction

Cancer continues to be the second leading cause of mortality worldwide and its prevalence is largely attributed to a lack of effective therapies in advanced, metastatic cancers, despite recent advances in immunotherapy and gene therapy [1]. The key challenges in developing new therapeutic protocols are controlled transport and localized drug delivery. Traditional therapy protocols are limited by the maximum allowable doses, precisely because of the severe side effects that can occur following treatment on normal cells, although higher doses would be necessary to counteract the rapid elimination and non-specific drug distribution to organs and tissues. However, this practice often leads to severe side effects, including the unintended destruction of healthy tissue. Moreover, many antitumor substances exhibit unfavorable physicochemical properties, such as reduced water solubility and stability, short half-life, and poor bioavailability. These factors require more frequent dosing intervals, which may compromise patient compliance and adherence to treatment [2,3].

There are new promising pathways to overcome those challenges while improving patient outcomes with the development of nanomedicines and innovative medication delivery methods, known as smart drug delivery systems. Biocompatible smart delivery systems remain a topic of maximum interest for researchers in recent years, with numerous chemotherapeutics being included in nanoparticulate matrices in order to reduce their cytotoxicity and improve their solubility and stability while ensuring a controlled and sustained delivery. Most anticancer substances have numerous drawbacks such as hydrophobicity, low water solubility, inadequate biodistribution, and susceptibility to multiple drug resistance, and their mere administration did not provide the desired efficacy, exhibiting dose-dependent side effects such as vomiting, hair loss, and nausea. This is the reason for their multi-cycle administration, in individualized therapies, varying the doses, frequency, and duration of each cycle. These drawbacks have urged researchers to find new modalities of transporting active substances using nanotechnology to enhance their pharmacokinetic and pharmacodynamic properties while achieving treatment effectiveness and reduced side effects [4,5,6,7,8,9,10].

Conventional drug delivery systems are based on the passive diffusion of antitumor drugs from vehicles, which means that the incorporated drugs are gradually and continuously released from the vehicles to a target site as well as into the bloodstream during their administration by simple diffusion. Therefore, the selective release of the loaded drugs cannot be completely controlled, resulting in suboptimal therapeutic efficacy. A solution to address this issue is the use of stimuli-responsive drug carrier systems, with drug release mechanisms that are triggered at the target sites. Stimuli-responsive biopolymers can respond to changes in environmental factors and endogenous and exogenous stimuli, including redox potential, pH, ions, temperature, light, electric field, and magnetic field [11,12,13,14] by self-changing either of their physical properties, chemical structures, or both. They can facilitate convenient drug loading and optimal drug release at the site of action [6]. These drug carriers offer important advantages such as high drug-loading efficiency, reduced drug toxicity, and more selective drug accumulation in tumors due to improved pharmacokinetics, superior biodegradability, and optimized renal excretion [4,5]. This represents a new drug delivery strategy for drugs used in cancer treatments that can provide maximal and selective therapeutic efficacy at the level of the target tumors, with locoregional drug accumulation, thus reducing unnecessary drug distribution in normal tissues, which will lead to the minimization of adverse effects. Drugs can be delivered to the target sites by passively targeting the carriers with a stimulatory response, subsequently released from those carriers by internal or external signals; such systems are referred to as “multi-targeting systems”.

Among the environmental factors that can stimulate the response of these stimuli-responsive drug carrier systems, we have discussed in this article folate-responsive delivery systems, glutathione- or redox-responsive delivery systems, and pH-responsive delivery systems. In recent years, to achieve selective antitumor drug delivery, folic acid (FA), a soluble derivative of vitamin B, and its receptor have been widely used as tumor markers. It is known that high-affinity folate receptors are overexpressed in various types of human cancers and undifferentiated metastases, whereas in normal tissues they have significantly low expression, thus giving tumor cells a growth advantage over neighboring normal tissue. These folate-responsive carrier systems showed an improvement in the antitumor activity of the delivered drug, explained by an increase in the speed and degree of their interaction with cancer cells, which resulted in a higher intracellular penetration of the administered drug [15]. On the other hand, stimuli-responsive drug carrier systems of glutathione (GSH) and pH are of particular interest because tumor microenvironments are mildly acidic and highly regenerative [16,17]. A balanced redox state is essential in many physiological processes such as proliferation, differentiation, cell death, gene expression, mitochondrial activity, Ca^2+^ regulation, immune response, and neuronal development, and therefore, changes in redox potential are very important. In healthy individuals, redox systems such as GSH and its disulfide (GSH/GSSG) function continuously to buffer reactive oxygen species (ROS) and maintain a stable environment. Patients with cancer, fibrosis, diabetes, and cardiovascular or neurological diseases have a dysregulated redox metabolism [13], and one of the natural biomarkers of cancer cells is the overexpression of GSH which acts as an antioxidant in inhibiting cell damage [17,18]. Drug delivery systems are also sensitive to pH function because the concentration of hydrogen ions varies according to the physiological compartment in which their release occurs [13,17]. As regards the environment in tumor tissue, the pH is usually acidic due to lactic acid produced by tumor cells through anaerobic glycolysis. The intercellular environment of the tumor has a pH between 6.5 and 6.8, and after endocytosis by the cancer cells, the pH of the endosomes/lysosomes is 6.0–4.0. The difference between the pH values can be used as a switch to trigger rapid disassembly of the antitumor drug-loaded system and at the same time, a large number of antitumor drugs could accumulate in tumor cells to enhance the therapeutic effect [18].

For the synthesis of these matrices, the use of biopolymers is the most common approach due to their biocompatibility with living tissues, non-toxicity, and biodegradability of their degradation products [4,5,6,11,12,19]. Numerous natural biopolymers are attractive for their design and there is an abundance of documented studies on this subject [10,20,21,22].

This review focuses on two key types of saccharides, chitosan and cyclodextrins, highlighting their diverse applications in pharmaceutical research, with a particular emphasis on cancer therapy. These biopolymers have been extensively studied due to their unique physicochemical properties, such as biocompatibility, biodegradability, and the ability to form complexes with various therapeutic agents. Chitosan is widely recognized for its antimicrobial, mucoadhesive, and drug delivery capabilities, while cyclodextrins are valued for their ability to enhance drug solubility and stability. Given their versatility and widespread use, we consider these two saccharides to be among the most significant representatives of oligosaccharides and polysaccharides in biomedical applications. We will describe the main properties of interest of the two targeted biopolymers, useful for the development of such nanosystems, and subsequently present their applicability across various cancer types.

When comparing chitosan- and cyclodextrin-based delivery systems to other established nanocarriers, several advantages and trade-offs become apparent. Liposomes, for instance, are clinically approved and well-characterized carriers that offer excellent biocompatibility and the ability to encapsulate both hydrophilic and hydrophobic drugs. However, they may suffer from stability issues and high production costs [23]. Dendrimers are highly tunable and can be precisely engineered for multivalent drug conjugation, but their synthesis is often complex and expensive, and toxicity concerns remain [24]. Polymeric micelles provide good solubilization of hydrophobic drugs and offer relatively straightforward preparation, yet their drug-loading capacity is often lower than that of cyclodextrin inclusion complexes [25]. In contrast, cyclodextrins enhance drug solubility through inclusion complex formation and are already used in several approved formulations [26], while chitosan offers mucoadhesive properties, inherent biocompatibility, and pH sensitivity that are highly beneficial for tumor-targeted delivery [27]. Together, these biopolymers offer an eco-friendly and cost-effective platform for smart drug delivery, particularly in stimuli-responsive and mucosal applications.

To set the stage for this review, it is important to briefly discuss how cyclodextrins (CDs) and chitosan (CS) stack up against other commonly used biopolymers in drug delivery. While our main focus here is on CDs and CS, other biopolymers like alginates, gelatin, and hyaluronic acid are also widely used in the pharmaceutical and biomedical fields. These biopolymers each have their own unique properties, such as differences in molecular weight, solubility, and biocompatibility, which can have a big impact on how well they work for delivering drugs and how suitable they are for clinical use. For example, CDs are especially useful for improving the solubility of drugs through inclusion complexes, while CS is known for its mucoadhesive properties and biodegradability, making it great for controlled, sustained drug release. On the other hand, other biopolymers might work better in specific therapeutic areas but can come with their own challenges, like stability issues or potential immune responses. By understanding what makes each of these biopolymers unique, we can obtain a clearer picture of the strengths of CDs and CS and how they can be applied effectively in different situations.

## 2. Characteristics of Biopolymers as Versatile Materials for Drug Delivery

### 2.1. Cyclodextrins (CDs)

Cyclodextrins (CDs) have been known since 1891 when they were isolated from starch or starch derivatives by Antoine Villiers using a specific enzyme—cyclodextrin-(CD)-glycosyltransferase. As early as 1953, CDs were patented for use in pharmaceuticals in order to improve the properties of active substances such as low water solubility or stability against oxidation [28]. Since then, interest in them arose considerably for both the pharmaceutical and food industries, because they are affordable and quite easy to produce. CDs are a class of macrocyclic oligosaccharides composed of 4–6 dextrose monomers linked by α-1,4-glycosidic bonds. The most important natural representatives include alpha-cyclodextrin (α-CD), beta-cyclodextrin (β-CD), and gamma-cyclodextrin (γ-CD) (Figure 1). Additionally, their synthetic derivatives refer to hydrophilic (2-hydroxypropyl-β-CD (HP-β-CD)), hydrophobic (2,6-di-O-ethyl-β-CD), and ionizable forms, such as sulfobutylether-β-CD (SBE-β-CD). Among those, HP-β-CD and SBE-β-CD exhibit the most favorable safety profile in biomedical uses and are included in parenteral formulations. Certainly, their toxicity must also be considered when administered by different routes of administration, as high doses can lead to irreversible renal failure, but in general, when administered orally, absorption is negligible due to their bulky and hydrophilic structure, rendering them as non-toxic components [29,30]. Native β-CD has certain limitations in use due to its very low aqueous solubility along with an adverse toxicological profile, characterized by nephrotoxicity and hemolytic activity. To address this concern, the modified hydroxyl group led to HP-β-CD, which was listed in US Pharmacopoeia and European Pharmacopoeia. This derivative has drawn significant attention due to its superior solubilizing capacity, comparatively greater inclusion-forming capabilities, and enhanced in vivo pharmacokinetic profile, rendering it more suitable for oral and intravenous administration than its parent β-CD [31].

Structurally, β-CD is characterized by the presence of a lipophilic central cavity and a hydrophilic outer surface, which are essential elements for establishing non-covalent bonds with various organic or inorganic compounds, thus inducing the formation of inclusion complexes that are particularly relevant in anticancer therapies, considering that most antineoplastic agents are hydrophobic [32].

The latest studies have demonstrated self-sufficient inhibitory activity for HP-β-CD on cell proliferation and metastasis in triple-negative breast cancer, specifically on 4T1, MCF-7, and MDA-MB-231 cells, highlighting its potential use as an antitumor agent. HP-β-CD has achieved therapeutic success being investigated for the treatment of Niemann–Pick disease, with phase III trials currently ongoing [33]. The mechanism of a presumed antitumor action lies in the depletion of cholesterol from cancer cells, with its accumulation at the tumor site being previously well documented [34]. Molecular perturbations in the expression of breast cancer-related genes in the cholesterol signaling pathway, analyzed via the RT2-PCR array, have identified the secreted frizzled-related protein 1 (SFRP1) as a directly correlated element with HP-β-CD using surface plasmon resonance drug–drug interaction analysis. That protein has an extremely high binding affinity to HP-β-CD [35]. The results showed cell growth inhibitory activity at a concentration of 10 mM HP-β-CD, confirming its capacity to interfere with mitochondrial processes, in agreement with previously published studies. The role of the SFRP1 protein as a tumor suppressor is often cited in the literature and consists of antagonizing the Wnt pathway by inhibiting Wnt-dependent transcription, leading to a cellular decrease in β-catenin, a Wnt activator. An impaired Wnt signaling pathway leads to cancer progression, stemness, treatment resistance, and immune escape [36,37,38].

Inclusion complexes with HP-β-CD manage to increase the in vitro solubility, dissolution rate, and oral bioavailability of various drugs, offering a favorable approach to improving their administration. Examples encompass the picoplatin-HP-β-CD inclusion complex in triple-negative breast cancer [39], alectinib-HP-β-CD [40], and abiraterone acetate-HP-β-CD for metastatic prostate cancer [31], among others. The therapeutic applications are limited by the lipophilic nature and low aqueous solubility for the active substances. All the aforementioned research confirmed a superior bioavailability for the inclusion complexes in comparison with the free drug.

CDs have a wide range of applications in the pharmaceutical industry, being extensively used to increase bioavailability, stability, or other pharmacokinetic properties of active substances due to their ability to form inclusion complexes. They are included in more than 40 drug formulations in the form of tablets, eye drops, and powders for oral suspension, and ointments in various therapeutic classes such as anti-inflammatory, analgesic, antipyretic, antitumor, etc. [41].

Nanomedicine aligns with the development of new drug delivery systems, and the design of such matrices is the fundamental strategy for future effective oncology therapies that will not only avoid the increased systemic toxicity of antineoplastic agents but will also facilitate their efficient internalization into tumor cells. In the past decades, numerous nanoparticulate biomaterial-based drug delivery systems have been designed to enhance therapeutic efficacy which coincides with the rapid progress in nanoscience; therefore, stimuli-responsive nanocarriers have been widely researched for effective targeted drug delivery [42].

The use of nanotechnology in the development of more effective delivery systems has materialized since the 1990s with the FDA approval of antitumor drugs such as Doxil^®^ (doxorubicin), DaunoXome^®^ (daunorubicin), and Abelcet^®^ (amphotericin B) as less toxic alternatives to conventional treatments, and more recently, for paclitaxel (Abraxane^®^, Nanoxel^®^, and Genexol-PM^®^) for the treatment of various types of cancers of the female reproductive system, lungs, and breast. Two doxorubicin (DOX)-liposomal formulations have been authorized for clinical use: poly(ethylene glycol) (PEG)-treated liposomal derivative (brand names: US: Doxil^®^, Lipodox^®^/generic Doxil/, Europe: Caelyx^®^) and non-PEGylated liposomal DOX (Myocet^®^) in Europe and Canada. The strategy of encapsulating DOX in liposomes is promising, but studies have shown that it does not provide superior efficacy in terms of progression-free survival and overall survival in comparison with conventional DOX. The safety profile is improved compared to free DOX, but PEG attachment may lead to mucosal and cutaneous toxicity, making administration dose-limited per injection. Preclinical studies have demonstrated improved efficacy when PEGylated liposomal DOX was functionalized with targeting ligands. However, the large size of antibodies that may restrict cellular penetration must also be considered. Consequently, other molecules such as aptamers, carbohydrates, peptides, and cell-penetration enhancers are currently being investigated as candidates for functionalization, either alone or in combination with other targeting ligands. Research in rodents has yielded encouraging results, but additional studies on the ideal ligand density and clearance of the functionalized PEGylated DOX require further exploration. Integrating certain stimuli, such as temperature or pH in the formulation of thermosensitive liposomes, or merging thermosensitivity with surface functionalization, may be an effective approach for DOX release [43].

Research into CD-based delivery systems, particularly β-CD and its derivatives, is expanding, encompassing NSAIDs (Flamexin^®^, Nimtech^®^, Bextrin^®^, etc.), antimycotics (Vfend^®^), antiarrhythmics (Nexterone^®^), or antipsychotics (Geodon^®^) [44].

The usual process of the controlled disintegration of CDs, and consequently drug release, is primarily attributed to pH fluctuations, which will result in hydrogen loss between the host and the encapsulated compounds. Their degradation may also result from heating or from the enzymatic cleavage of α-1,4 links between glucose units. CD complexation can yield advantages, such as differentiated release for multiple guest molecules which may be a desired outcome following the alteration of the initial formulation. This is attributable to the varying solubility of each complex and the variation between their release rates. As we will comprehensively detail in the subsequent studies, the functionalization of CDs is one of the most common approaches in developing smart drug systems. Various ligands can be attached to CDs in order to selectively target receptors that are overexpressed at the tumor site (e.g., folate receptor, biotin receptor, and glucose receptor) [45].

There are two main approaches to antitumor drug delivery systems: active and passive targeting. Active targeting focuses on a specific peptide, antibody, or fragment of an antigen or receptor overexpressed within the tumor, whereas passive targeting utilizes the enhanced permeability and retention (EPR) effect that arises from the tumor’s insufficient lymphatic system and the scarcity of lymphatic capillaries, as it is represented in Figure 2. That type of targeting is used in formulations like liposomes, nanoparticles (NPs), or macromolecular transport systems [46].

The stability and solubility of numerous antitumor drugs have been enhanced through CD-inclusion complexes, leading to a much improved pharmacological profile and, ultimately, therapeutic success [48,49]. Therefore, bioresponsive types of CDs for specific and sustained release of encapsulated drugs have been designed. Below, we present a few examples based on research conducted in recent years.

### 2.2. Chitosan (CS)

Chitosan (CS) is a biopolymer frequently employed as a smart drug delivery system across various active pharmacological substances. Similarly to CDs, CS possesses specific benefits that make it a perfect fit for biomedical applications, being non-toxic, non-immunogenic, biodegradable, and stable under physiological conditions. It possesses both mucoadhesive and absorption-enhancing properties, key elements in its use as a smart drug delivery system. Consequently, its use in nanomedicine is justified, as it can mitigate the limitations of certain drugs in terms of their pharmacokinetic, pharmacodynamic, and toxicological properties. It has a high potential for drug and gene delivery, cell culture, and tissue engineering, along with several individual characteristics, including wound healing and antimicrobial and antifungal activity [50,51,52,53].

CS (Figure 3) is a chitin derivative, obtained by partial deacetylation in an alkaline environment, with a molecular weight varying from 50 to 1000 kD and a degree of deacetylation of 30–95% depending on its source. It is inherently found in the shells of crustaceans, crabs, and shrimps; fungal cell walls; and the exoskeletons of insects. Structurally, it is a polysaccharide consisting of deacetylated D-glucosamine and acetylated N-acetyl-D-glucosamine monomers, interconnected by β-(1→4) linkages. CS dissolves in slightly acidic aqueous solutions and after neutralization, the CS transforms into hydrogels [54].

According to the study conducted by Lim and coworkers [55], the degree of acetylation significantly influences the physical, chemical, and biological properties of CS, which have major implications for its medical applications. Their results indicated that the modulation of the degree of acetylation affects the adhesive properties of CS, as high molecular weight CS exhibits a strong adhesion and cohesion due to the chain’s flexibility, which diminishes as the degree of acetylation values increases. CS has a polycationic nature that contributes to its bioadhesive properties and solubility. It can easily interact with negatively charged surfaces such as mucous membranes, thus extending the contact time with the encapsulated active drug [50].

Despite its attractive features, CS has a weak mechanical and fast degradation rate, limitations that may be overcome by pairing it with other polymers. There are specific reactions involving the –NH2 group of C-2 or nonspecific reactions of the –OH groups of C-3 and C-6 that are commonly implemented to synthesize CS derivatives such as O– and N-carboxymethylchitosans, chitosan-6-O-sulfate, trimethylchitosan ammonium, trimethylchitosan ammonium, chitosan-grafted copolymers (notably poly(ethylene glycol) (PEG)-grafted CS), alkylated CS, and CD-linked CS. The main strategies include crosslinking, graft copolymerization, carboxymethylation, etherification, and esterification in order to functionalize the CS’s structure [56,57].

CS can be employed as an effective excipient for the controlled release of various drugs, forming CS-based systems such as microspheres, nanoparticles, hydrogels, capsules, etc. Numerous studies have shown promising results regarding embedding antitumor drugs in such a matrix, with good biocompatibility with healthy tissues, CS being an excellent carrier for those drugs.

There are several pathophysiological particularities of the tumor microenvironment (TME) that may have significant potential to be exploited to develop drug delivery systems that allow drug release at the tumor site. The pharmacokinetic characteristics, tumor accumulation efficacy, and toxicological profiles of current pharmaceuticals can be effectively optimized by the pH value, folate receptors, and glutathione level.

If we consider pH-responsive drug delivery systems and that acidosis is a universal feature of TME, conducting the development of medicines that respond to this stimulus could be successful. TME is characterized by an acidic pH between 6.0 and 6.5, and chitosan undergoes protonation and solubilization to a greater extent in acidic environments; thus, chitosan molecules are well dispersed through electrostatic repulsion and transit from a tightly coiled to an uncoiled elongated linear configuration, which can successfully interact with the negatively charged membranes of cancer cells and endothelial cells of tumor vasculature, which overexpress anionic surface moieties such as phospholipids, glycoproteins, and proteoglycans. This allows chitosan to affect the cell membrane permeability and migrate intracellularly to exert anticancer effects including anti-metastasis via suppressing the production of certain enzymes such as matrix metallopeptidase 9. Conversely, at neutral physiological pH, chitosan exhibits a lower affinity towards the normal cells which concludes its selective feature for cancer cell targeting [58].

The molecular weight (MW) and degree of deacetylation (DD) are critical factors influencing the application of chitosan as a drug delivery matrix, as they directly affect both drug encapsulation efficiency and drug release kinetics [59].

In the conducted studies, chitosan with a medium degree of deacetylation and very low molecular weight was the most commonly employed. It was observed that formulations with higher DD exhibited greater entrapment efficiency (EE) of pharmaceutical compounds [60]. This suggests that chitosan with a high DD contains a greater number of free amino groups, which enhance interactions with active drug molecules via hydrogen bonding [61]. Conversely, chitosan with a lower DD resulted in reduced drug-loading efficiency [60].

Regarding molecular weight, systems utilizing low MW chitosan demonstrated superior encapsulation performance, whereas formulations incorporating medium and high MW chitosan exhibited a diminished capacity for drug encapsulation. This phenomenon is likely attributable to the increased viscosity of high MW chitosan solutions, which can hinder the accessibility of amino groups for drug interaction in more viscous environments [62].

The influence of MW and DD on controlled and sustained drug release has also been extensively examined [63,64,65]. It was found that drug release is most effectively prolonged in the presence of chitosan with both high MW and high DD. The interaction between chitosan and the encapsulated drug plays a crucial role in modulating the release profile, with these interactions becoming stronger as MW and DD increase [66]. In addition to the degree of deacetylation and molecular weight, other structural parameters of chitosan can also influence drug-loading capacity. These include the polydispersity index, which reflects the distribution of chain lengths, and the chain flexibility that may impact the entanglement and interaction with guest molecules. Moreover, the molecular conformation of chitosan in solution, which depends on pH and ionic strength, can modulate the formation of nanoscale structures with cyclodextrins. The electrostatic interaction between the protonated amino groups of chitosan and negatively charged drugs, as well as the size compatibility with the cyclodextrin cavity, are also crucial aspects that determine drug encapsulation efficiency and release dynamics [67].

Combining chitosan and cyclodextrin in drug delivery systems has garnered significant attention in cancer therapy due to their complementary properties. Chitosan offers biocompatibility and mucoadhesive characteristics, while cyclodextrins enhance drug solubility and stability. Recent studies have developed innovative delivery vehicles utilizing these materials to improve therapeutic outcomes in cancer treatment [68].

## 3. Stimuli-Responsive Polymers Based on Cyclodextrins (CDs)

### 3.1. Folate-Responsive CD-Based Delivery Systems

A diverse array of cell targeting agents such as aptamers, antibodies, hyaluronic acid, and folic acid (FA) can be used for selective transport to specific targeted cells. Those targeting agents can bind to their specific receptors on the surface of cancer cells, thereby promoting the internalization of encapsulated therapeutic agents. One such example is the extensively studied FA, which binds to its specific folate receptors (FRs), a type of membrane-associated protein. The intended outcome is the reduced systemic distribution of the drug, resulting in an increased toxic effect on tumors and decreased toxicity on healthy cells. Therefore, patient compliance with treatment will also be improved as a result of the administration of those nanocarriers [69,70].

Folate-receptor drug delivery refers to drug release based on the quantity of folate receptors present on tumor cells.

Folate is an essential water-soluble vitamin, necessary for growing cells. Considering that cancer cells are rapidly dividing, they require a substantial supply of folate in order to fulfill their metabolic requirements. Folate receptor alpha (FRα) is a membrane-bound protein that exhibits a high affinity for folate and facilitates its transport. It is overexpressed in various malignancies of the ovary, breast, pleura, lung, cervix, endometrial, kidney, bladder, and brain. Notably, FRα expression in normal tissues is restricted only to the luminal side of the kidney and lungs, and, with the exception of the kidneys, FRα is not accessible to folate in the bloodstream. Therefore, numerous variants of folate-conjugated nanosystems and antifolates have been developed to target FRα-positive tumors [71]. Figure 4 summarizes a range of FRα-targeting approaches for cancer therapy including chemotherapeutic conjugates, vaccines, T-cell therapies, and monoclonal antibodies for clinical application.

An ideal transport system for antitumor drugs would entail its ability to transport the desired therapeutic agent to the target while maintaining its stability. The ultimate goal is tumor-specific release and accumulation, for which the variant of conjugating an entity to the surface of NPs may be a promising strategy. Considering that FR is overexpressed on the surface of more than 40% of solid tumors, including ovarian, breast, renal, etc., the integration of FA into a β-CD or hyaluronic acid-based copolymer coated with poly(ethylene glycol) PEG for active targeting represents a feasible solution [73,74,75,76,77].

Thus, Hong et al. developed a nanodrug system (FA-Cur-NPs) utilizing folate-conjugated β-CD-polycaprolactone block copolymers composed of β-CD and ε-caprolactone (ε-CL) for rapid intracellular release after the receptor-mediated endocytosis of curcumin [78]. FA-Cur-NPs were 150 nm spherical particles, with an encapsulation efficiency (EE%) higher than 20%, and a dual-responsive behavior that is temperature- and pH-dependent. Thus, at pH = 6.4, curcumin release occurred three times faster, which was explained by the acidic pH of tumors at which ester bonds are severed. The intracellular uptake of the FA-Cur-NPs system was superior to Cur-NPs in both in vivo and in vitro studies due to folate receptor-mediated endocytosis. Cell viability was assessed by MTT assay on the L929 (fibroblast) and HeLA (human cervical cancer) cell lines, and in vivo antitumor activity was demonstrated on the HeLA xenograft mice animal model. The study demonstrated a promising approach to improving the current anticancer therapies through active targeting and controlled release.

Immunotherapy in current cancer management is expanding rapidly, and an important branch is represented by immune checkpoint inhibitors.

Remodeling the inherent immunosuppressive nature of the TME to induce immunogenic cell death can be achieved using chemotherapeutic agents. That perspective was explored by Sun et al. in their study evaluating ginsenosides from ginseng extract, specifically the potential of ginsenoside RG3 as an inducer of immunogenic cell death in combination with quercetin (QTN) capable of releasing ROS [79,80]. Those active substances were simultaneously encapsulated in a carrier system targeting folate in the form of amphiphilic PEG-coated β-CD-based NPs. The obtained nanoparticulate system had a promising EE% higher than 95%, and the cytotoxicity, antiproliferative, and antimetastatic activities of the two co-encapsulated substances were evaluated by the MTT method on the CT26 and HCT116 colorectal cancer cell lines. The results confirmed the induction of immunogenic death in tumor cells, demonstrating the synergistic anticancer effect of the two encapsulated substances, targeting folate receptors and altering the immunosuppressive character of TME when combined with therapy-programmed death-ligand 1 (PD-L1). The FA-targeted coformulation significantly increased the survival of tumor-bearing animals in an orthotopic colorectal tumor mouse model by reversing the immunosuppressive effect of the TME.

### 3.2. Glutathione (Redox-Responsive) CD-Based Delivery Systems

Another stimulus related to the pathophysiological environment of tumor cells to which nanoparticulate systems can respond for the rapid and targeted delivery of encapsulated drugs is the level of GSH, an intrinsic redox-intensive system used in the development of stimuli-responsive nanocarriers. Specific systems have been developed that rapidly and aggressively deliver the active substance to the tumor, with significantly diminished side effects. A first example in that category are polymeric micelles which possess several unique attributes, such as improved intratumor retention through the EPR effect, prolonged in vivo circulation time, enhanced bioavailability of poorly soluble drugs, increased therapeutic efficacy, and fewer side effects, to mention a few.

The release rate of the active substance was proportional to its diffusion rate or the decomposition rate of the polymeric micelles in which it was embedded. Specifically, in an environment with a high concentration of GSH, the decomposition of micellar systems could lead to an accelerated release or to an increased diffusion of the encapsulated drug. Therefore, the development of polymeric carrier systems that include properties responsive to such stimuli (GSH, pH, temperature, etc.) should lead to more effective treatments [81].

The mechanism for site-specific drug delivery and release in response to redox gradient lies in the difference in GSH concentration which is higher in tumor cells (0.5–10 mM) compared to the normal cells (2–20 µM). The presence of disulfide groups in the drug delivery systems promotes the release of drug molecules in response to the high intracellular GSH concentrations [82]. Using that mechanism, Li et al. [83] synthesized an amphiphilic copolymer based on PEG and poly(ε-caprolactone) (PCL), β-CD-g-PCL-SS-PEG-FA, a unique structure that could self-assemble into a multifunctional micelle with excellent stability in aqueous environment. Figure 5 illustrates the design of those unimolecular micelles followed by their targeted delivery and enhanced therapeutic efficacy against multidrug-resistant (MDR) cancer cells.

The micelles displayed redox properties due to the presence of disulfide groups at the junction between PEG and PCL while also targeting the same FRs previously mentioned. The FRs located on the outside of the micelles were transport vectors to the tumor target cells, following the PEG-coated hydrophilic core detachment due to the increased intratumoral level of GSH, thus becoming a trigger in cell cytoplasm, resulting in rapid release. Their study focused on doxorubicin (DOX), a drug with an unfavorable pharmacotoxicological profile when administered either by itself or when loaded in non-FA-containing carrier systems. The DOX-loaded nanoparticulate system was characterized and evaluated for cell viability and antitumor activity by MTT assay on the HeLA/MDR1 and HepG2/MDR1 cell lines. The drug loading percentage was 8.1%, and the particle size was 31.5 nm. The increased intracellular uptake as well as inhibition of tumor development could have been due to overexpressed FRs inducing the reuptake of FA-containing compounds into the intracellular medium with increased GSH level, causing immediate DOX release. The results demonstrated good biocompatibility and low in vitro toxicity. All the data highlighted the potential of that system for the management of cancers resistant to conventional treatments.

DOX has also been analyzed in a study led by Daga et al. [84]. It was embedded in a β-CD-based nanosponge (NS) nanoparticulate form, responsive to high intratumoral GSH levels, aiming at controlled release, increased water solubility, bioavailability, and stability of the incorporated active substance. Its structure is displayed in Figure 6. Structurally, those types of formulations were cage-like matrices obtained via crosslinking of α, β, and γ-CDs with proper crosslinking agents creating nanochannels in the polymer matrix, that could encapsulate a large variety of active substances. In addition, the hepatotoxicity of both the free and polymerized forms was evaluated in vivo and ex vivo. Elevated GSH levels served as a specific marker for chemoresistant cells, and antitumor efficacy correlated directly with those levels [85]. With those theoretical considerations in mind, the authors developed a GSH-responsive polymer based on β-CD and loaded it with DOX and 6-coumarin. The two active substances were simultaneously incorporated into the polymer matrix by mixing with aqueous GSH-NS suspension. The major advantage of that system was the dual responsiveness to both stimuli, pH and GSH, that trigger drug release. Cell viability compared to the free form was evaluated on the HepG2 cell line, the hepatocarcinoma cell model, demonstrating similar inhibitory capacity, hence safety in administration. However, the high level of DOX in the particulate forms in DOX-GSH-NS treated cells was due to the intracellular accumulation of NS, also demonstrated by rapid fluorescence (15 min) in 6-coumarin-GSH-NS treated cells. The ex vivo confocal microscopy study on rat liver tissue confirmed the accumulation of the DOX-GSH-NS-loaded system to the same extent as the free form of DOX. The safety profile was found to be within acceptable parameters, exhibiting comparable toxicity and being able to bypass the P-gp efflux pump, thus contributing to counteracting treatment resistance. Moreover, the unloaded GSH-NS system did not cause any in vivo or ex vivo toxicity and could be successfully used for the incorporation of other therapeutic agents. Promising results for that DOX-loaded system can also be seen in a previous study by the same authors in a prostate cancer cell model in which not only its efficacy but also the lack of DOX cardiotoxicity compared to the free form was demonstrated [86].

Subsequently, the biological effects of that type of system were evaluated in vitro in a study led by Argenziano et al. [87]. Two GSH-NS-type formulations with various disulfide bond contents (GSH-NS B with low disulfide bond content and GSH-NS D with high disulfide bond content) were synthesized. Their biological effects on cell division, mitochondrial activity, membrane integrity, mRNA expression, and ROS production were analyzed on colorectal cancer (HCT116 and HT-29) and prostate cancer (DU-145 and PC3) cell cultures, with the highest concentration of GSH observed in HCT116 and DU145. That type of polymer system, NS, was formed by crosslinking α, β, or γ-CD using as crosslinking agents carbonyl compounds such as carbonyl-diimidazole, diphenylcarbonate, or organic dianhydrides such as the pyromellitic dianhydride. Previous in vivo studies have shown no toxicity after cellular exposure to uncharged NS systems at concentrations ranging from 10 to 100 μg/mL. In addition, the use of pyromellitic dianhydride as a crosslinking agent has been shown to be safe upon oral administration without toxic side effects. Their potential for use as smart delivery systems for various antitumor agents was demonstrated in that study, with cell cycle blockade only occurring at concentrations significantly higher than those used in delivery systems [88,89]. Concerning the impact on cell proliferation in the specified tumor lines, the most susceptible to the cytotoxic action of GSH-NS D was the DU145 line, probably due to its higher GSH content. HCT116 colorectal cancer cells had a pronounced effect on the cellular reuptake of GSH-NS B and D compared to the other cell lines; the differences were due to thiol groups, the cleavage of disulfide bridges starting from the surface by thiol/disulfide group exchange reactions catalyzed by redox proteins such as thioredoxins. Moreover, due to the significant cellular reuptake of GSH-NS, considerable ROS production also occurred after the exposure of the HCT116 line to those systems. In a cell cycle analysis, a significant decrease in the G0/G1 phase was observed in all the tumor lines within 24 h at IC50 values previously evaluated in the study of GSH NS systems. To investigate the blockage of that cycle, some genes involved in the process were analyzed. The downregulation for mRNA of cyclin CDK1, CDK2, and CDK4 and upregulation on the inhibitory genes CDKN1A and CDKN2A were observed, results which were in agreement with those obtained by Choi et al. in their study. The cytotoxicity of the systems analyzed correlated with the number of disulfide groups, with GSH-NS D being the least toxic in all the cells except DU145 [90].

The generation of ROS is a fundamental approach in anticancer therapy. The objective is to influence the equilibrium of ROS formation and ROS removal at the cellular level, as those entities can damage cells, DNA, proteins, and lipids when present in excess. Therefore, altering redox homeostasis is a promising strategy in the development of novel oncologic therapies. Among them, Xu et al. [91] explored ferroptosis, a novel non-apoptotic cell destruction mode that relied on the characteristics of the ROS-dependent iron/lipid peroxidation system. The authors synthesized a ROS-generating nanoparticulate system loaded with dihydroartemisinin (DHA), coated with a pH-sensitive ferrous polyphenolic network. More specifically, NPs were obtained from acetalated β-CD, subsequently coated with a Fe^3+^-polyphenol (tannic acid) network in which the active substance was embedded. The obtained system was able to prevent the rapid leakage of the drug into systemic circulation by efficiently protecting the functional groups in the DHA structure and preventing the premature loss of drug cargo by exploiting the intratumoral acidic pH at which the system disintegrated and released the drug and ferric ions. High concentrations of GSH reduced Fe^3+^ to Fe^2+^, which enhanced the iron ion-dependent cytotoxicity of DHA.

Supramolecular design in anticancer therapy is a novel approach that relies on drugs as building blocks with clinical applicability, with benefits including the controlled release of multiple and varied components, responsiveness in intratumoral settings, high therapeutic efficacy, and minimal side effects [92]. One such supramolecular redox-responsive nanoparticulate system was synthesized by Zhang et al. [93] in which they aimed to encompass cisplatin, a drug with restricted use in therapy due to its pronounced systemic toxicity and resistance to treatment. That system, namely Pt(IV)-SSNPs, was developed by the complexation of β-CD polymer with a modified adamanthyl-platinum IV prodrug, Pt(IV)-ADA2. It was proven to be capable of reduction to the active cisplatin form in a strongly reducing environment, such as that of tumor cells, which subsequently reduced its systemic toxicity. Antitumor activity was demonstrated both in vitro and in vivo in mice, with the ability to inhibit tumor development at different stages. Histologic analysis demonstrated that the system developed by the authors was most effective in inducing cell apoptosis and necrosis.

### 3.3. pH-Modulated Drug Release from CD-Based Delivery Systems

5-fluorouracil (5-FU) is a fluoropyrimidine antimetabolite that interferes with DNA and RNA normal functions, replacing the uracil base in their structure [94]. A major disadvantage of 5-FU therapy is the unfavorable pharmacotoxicological profile, affecting hematologic, gastrointestinal, and bone marrow structures [95,96,97]. To counteract its toxicity and its rapid distribution, characterized by a short half-life of 6–22 min with modest permeability, the integration of 5-FU in nanomedicines, either in monotherapy or alongside other chemotherapeutics, has been the subject of extensive research in recent years.

Akkın et al. [98,99] examined the inclusion complexes of 5-FU in novel formulations featuring CDs as a potential treatment for colorectal cancer. The researchers evaluated the possible synergistic effects of 5-FU and interleukin-2 (IL-2) by developing a nanoplex employing a cationic CD carrier and epichlorohydrin-βCD (EPI-β-CD) as a crosslinker, which prior studies have demonstrated to enhance drug absorption and bioavailability [100]. IL-2 is an essential cytokine in immunotherapy, facilitating the activation of T cells that support the immune system’s long-term response and memory. Nevertheless, it has the drawback of a short half-life of merely 7 min, requiring frequent high-dose administration to sustain efficacy. As a result, there is a potential for severe side effects, including neutrophil dysfunction and capillary leak syndrome [101]. To mitigate those side effects, various research examining the combination of IL-2 with radiation, paclitaxel, cyclophosphamide, or the PEGylated version of nivolumab have shown enhanced antitumor efficacy in preclinical studies [102,103]. In the aforementioned study, three 5-FU/IL-2/CD nanoplexes were synthesized utilizing the interaction between cationic CD polymers and IL-2 encapsulated with 5-FU to achieve controlled release in colon tumors. The drug loading and EE% results were approximately 40% for 5-FU and 99.8% for IL-2, consistent with prior studies. 5-FU has been previously incorporated into several CS-based matrices, including nanoparticulate forms, achieving EE% values of 44.2%, 48% in a co-loaded form with curcumin, and 55% when combined with doxorubicin using the same biopolymer [104,105,106]. On the other hand, the authors were able to achieve almost complete EE% for IL-2 (99.8%). While the acquired data require supplementation through in vivo studies, the in vitro findings have revealed the significant potential of those CD-based nanoplexes for the delivery of combined pharmacological agents, indicating applicability for various administration routes (parenteral or oral) in the management of colorectal cancer.

Recent research has extensively investigated the application of epichlorohydrin (EPI) as a crosslinker for the enhancement of drug delivery systems. The EPI-β-CD polymer is an effective carrier for specific molecules, enhancing their solubility, bioavailability, and, thus, therapeutic efficacy. That β-CD-derived polymer integrates the benefits of water solubility and high molecular weight typical of polymers with the capacity to generate inclusion complexes specific to CDs [107,108]. Moreover, in many studies, the option of polymerizing β-CD with EPI was evaluated in order to form a stable self-assembling hydrogel able to incorporate hydrophobic molecules, which was impossible with native β-CD, which presents the disadvantage of low water solubility and low molecular weight. The covalent grafting of CS to β-CD under mild conditions using EPI as a crosslinker leads to a highly hydrophilic porous network wherein the release of the encapsulated drug is sensitive to pH and temperature variations. Figure 7 illustrates the successive reactions required to synthesize that polymer.

Multiple studies have focused on the synergistic pharmaceutical effect of two drugs, as demonstrated by Almawash and colleagues in their research on 5-FU and methotrexate (MTX) [110]. The crosslinking approach utilizing epichlorohydrin sought to extend the in vivo release of 5-FU and MTX. Using it as a crosslinker offers numerous intrinsic advantages, including biocompatibility and the absence of harmful side effects. Nanosystems crosslinked with epichlorohydrin are among the most efficient drug delivery systems that can be used for targeting 5-FU in colon cancer. It is also known as a bifunctional mutagenic alkylating agent, and it is being used in the formulation of drug carrier systems composed of mucilages [111].

Both exhibit significant toxicity as standalone therapies due to their low selectivity in contact with healthy tissues and can lead to MDR. Consequently, they are frequently included in formulations such as hydrogels [112], microemulsions [113], microparticles [114], nanoparticles [115,116,117], implants [118], etc., to facilitate a more targeted delivery to tumors, an improved efficacy, and to reduce side effects. The aforementioned study sought to incorporate the two drugs into self-assembling hydrogel formulations for subcutaneous injection. Those were synthesized by inclusion complexation between cyclodextrin-derived host-type polymers capped with cholesterol and adamantane as guest-type polymers. Two hydrogels were loaded with 5-FU and MTX and evaluated for morphology using SEM analysis, rheological behavior, in vitro release profile, cytotoxicity against MCF-7 breast tumor cell line via MTT assay, and histopathological alterations following intratumor injection. Following the validation of a pH value that was suitable for subcutaneous injection, ranging from 7.2 to 7.4 without causing irritation or damage to human tissues, the antitumor activity confirmed the therapeutic efficacy of the two formulations. The combination treatment exhibited a significant cellular inhibitory effect and an improved antiproliferative capacity versus monotherapy. The absence of toxicity in empty hydrogels on both the normal and cancer cells rendered them promising candidates as transport vectors for other active compounds. A final histopathological analysis confirmed the decrease in tumor manifestations, with the formulated hydrogels serving as optimal matrices for the 5-FU/MTX combination that was delivered in an efficient and safe manner, which could be extended to other antitumor substances.

Another study conducted by Zhao and coworkers [119] included the same combination of active substances, as 5-FU and MTX conjugate in a sulfobutyl ether-β-cyclodextrin complex, also designed for injectable administration. Employing that technique, the authors obtained a 92-fold enhancement in the solubility of 5-FU and MTX conjugate and a two-fold extension of the half-life of 5-FU. The ester-like structure of the conjugate was stable and safe under physiological conditions and was subsequently hydrolyzed under the action of hydrolases resulting in the controlled release of the two active drugs. In vivo experiments carried out by the authors were performed on the MC38 mouse ectopic colon cancer model to assess the antitumor activity of the proposed conjugate in comparison to a 5-FU and MTX solution, demonstrating enhanced effectiveness likely attributable to the increased half-life and slower metabolism of the two individual components.

The same crosslinker, EPI, was also used in Giglio and coworkers’ study for the incorporation of sorafenib (SFN), a BRAF V600 inhibitor tumor agent [120]. SFN inclusion complexes, referred to as pCyD/SFN and oCyD/SFN, were analyzed for their antiproliferative effects on the following cell lines: MCF7, HGC-27, HepG2, SKMel-28, and K1 cancer cell lines in comparison to free SFN. The synthesized inclusion complexes were able to trigger cell apoptosis with a lower in vivo toxicity compared to the free drug. In vivo, experiments conducted on tumor-bearing mice and histological analysis of the main organs indicated a reduced overall toxicity and, in particular, almost no liver toxicity compared to free SFN. Tumor selectivity can be enhanced by increasing the affinity for folate receptors, as demonstrated by the same authors in their study incorporating in a crosslinked CD polymer the antitumor agent LA-12 (cis-trans-cis-[PtCl_2_(CH_3_CO_2_)_2_(adamantlyNH_2_)(NH_3_)]), whose cytotoxicity on the MDA-MB-231 cell line increased following the FA functionalization of that polymer [121].

The complexation of 5-FU with β-CD or HP-β-CD represents an advantageous formulation approach favoring controlled and complete release with superior efficacy when administered as a heat-sensitive mucoadhesive gel as demonstrated by Bilensoy et al. in their study [122]. The cytotoxicity of the formulated gel in this study was assessed using the MTT assay on the HeLa An1 human cervical carcinoma cell line, indicating the potential gel’s safety upon topical administration. Cytotoxic effects were observed at significantly reduced doses, resulting in an antitumor activity up to ten-fold greater due to the complexation of 5-FU in CDs, as confirmed by the authors and supported by an enhanced release profile. The advantage of that gel was the prolonged vaginal retention. The indications could be extended to genital warts with dose adjustment.

The abovementioned studies are presented in Table 1 in a more concise manner.

## 4. Stimuli-Responsive Polymers Based on Chitosan (CS)

### 4.1. Folate-Responsive CS-Based Delivery Systems

Folic acid (FA), known as vitamin B9, is an essential micronutrient involved in the metabolism of nucleic acids (DNA and RNA). Its drawback is that it undergoes an intense degradation process as a result of light, pH variations, and certain oxidative processes, raising stability issues. Even upon oral administration, its bioavailability is limited due to diminished absorption and structural breakdown. In order to limit those challenges, encapsulation strategies employing biopolymers have been developed as a particularly efficient approach. Among the natural polymers, CS is considered to be ideal and extensively utilized in the pharmaceutical and food industry [123,124,125].

Folate-engineered polymers are new nanocarriers that show great potential in delivering anticancer drugs at the tumor sites, reducing toxicity and possible drug resistance. There are several types of cancer cells (fallopian tube, epithelial tumors of the ovary, and primary peritoneum; lung, kidney, ependymal brain, uterus, breast, colon, and malignant pleural mesothelioma) that exhibit a high density of outer surface folate receptors, making folate-drug conjugate-incorporated NPs a particularly useful strategy in oncologic therapies [47,126,127]. The release mechanism of the active substance from such polymers is shown schematically in Figure 8.

Regarding therapeutics, multiple antitumor substances have been researched using that medical approach. One such example is sorafenib (SFN), a kinase inhibitor, previously mentioned in an inclusion complex with CDs [120]. In the study conducted by Narmani et al. [61], that drug was included in a novel FA-functionalized PLGA-CS smart delivery system. Due to that, drawbacks such as low hydrophilicity, weak bioavailability, and solubility were counteracted while inducing apoptosis in cancer cells and repressing their proliferation differentiation, growth, and angiogenesis in cancer tissues, as well as blocking serine/threonine kinases [127,128]. The nanoparticulate matrix proposed by the authors has been evaluated by various biomedical assays such as cell cycle arrest, cell apoptosis, and uptake assays for the confirmation of antitumor action on A549 lung cancer cells.

In a study conducted by Abdipour et al. [129], multifunctional microbubbles comprising poly(lactic-co-glycolic acid), CS, polyethylene glycol, and FA were evaluated for their controlled release of MTX. The copolymer was synthesized for targeted drug delivery due to the overexpression of folate receptors on the surface of cancer cells by conjugation of FA to the previously prepared PLGA-CS-PEG MBs. The novel polymeric formulation incorporated gold nanoparticles as a therapeutic enhancer of the widely used anticancer drug, MTX in a perfluorohexane core. After a thorough characterization of the synthesized materials, confirming its structure by FTIR and ^1^H NMR analysis, key parameters such as particle size, morphology, responsiveness to ultrasound and temperature, encapsulation efficiency, release profiles, and in vitro cytotoxicity tests were employed. In agreement with prior research, that study also illustrated higher internalization of MTX due to receptor-mediated opsonization [130]. The active targeting ability of the FA-nanodroplets with ultrasound radiation was confirmed by flow cytometry and the apoptotic properties of MTX-loaded nanodroplets were investigated via DAPI staining assay and MTT assay on the MCF-7 breast cancer cell line. The frequency of chromatin fragmentation in MTX-loaded nanodroplets and MTX-loaded FA nanodroplets, both in the presence of ultrasound irradiation, was more than that of free MTX. Therefore, that copolymer could be an excellent nanocarrier in cancer therapy applications.

The cytotoxic activity on MCF-7 cells of another newly synthesized polymer responsive to folate receptors was evaluated by Geethakumari et al. [131]. The researchers have evaluated the possibility of incorporating cytarabine in a nanoparticulate polymer to counteract its adverse effects and potentiate its effect on solid tumors. The cytarabine-containing folate-conjugated CS nanoparticles (FCCNPs) were prepared by ionic gelation using tripolyphosphate (TPP) as a crosslinker. Their average diameter was 50 nm, a suitable size for targeting cancer cells. In addition to previous studies, the authors have evaluated the relative expression of *bax*, *cyt c*, and *cas 9* apoptotic genes after the treatment with the loaded copolymer. The results were promising, as their expression increased after 72 h, inducing apoptosis and inhibiting tumor cell proliferation, as shown in Figure 9.

In another study, irinotecan (IRI) and quercetin (QTN) were co-encapsulated in a polymeric nanoparticle system for colon cancer treatment [133]. The use of both active substances was based on their synergistic effect, which reduced dosage and minimized multidrug resistance while maintaining drug efficiency [134]. The co-encapsulation of the two active substances in the form of folic acid–CS-modified PLGA nanoparticles (C-FA-PNPs) has shown therapeutic potential during in vitro studies. Previous research has confirmed that colon tumor cells exhibit an overexpression of folate receptors, thus validating the synthesis of nanoparticles reported by the authors in their study. Considering the negative charge of PLGA on the surface, which would limit the interaction with similarly negatively charged surface cells, the intracellular uptake of the active principles would be significantly reduced. Therefore, a CS and folic acid coating was proposed in the matrix design to precisely target tumor cells [135,136,137]. After the physicochemical characterization of the synthesized NPs which confirmed the incorporation of the two analyzed substances, in vitro studies evaluated their release profile as well as their cytotoxicity on the Caco-2 cell line. The controlled release from the C-FA-PNP matrix for the two active substances was due to the distinctive properties of CS which protonated at low pH and led to increased permeability. When simulating the acidic pH of gastric media, a 2 h release of up to 46.59 ± 3.48% (IRI) and 34.3 ± 2.81 (QTN) was observed while at intestinal pH = 6.8, it was observed that approximately 16.41% of IRI and 8.8% of QTN was released. Cytotoxicity studies conducted on the Caco-2 cell line have demonstrated lower viability for the nanoparticulate formulation containing both substances, suggesting promising therapeutic potential, further supported by in vivo studies on colon cancer-induced rats.

Biocompatibility and safety are crucial factors in selecting polymers for drug delivery applications. Among biodegradable polymers, CS and PLGA are widely used due to their favorable drug-carrying properties and established biomedical applications. However, their cytotoxicity profiles differ significantly. CS is generally recognized for its biocompatibility and biodegradability, though its molecular weight and degree of deacetylation can influence cellular responses. Some studies have reported that high concentrations or low molecular weight CS may induce cytotoxic effects in certain normal cell lines due to membrane interactions and ROS generation [138]. On the other hand, PLGA, an FDA-approved polymer, is extensively used in clinical formulations due to its well-documented safety profile. However, its degradation products—lactic and glycolic acid—can lead to localized acidity, potentially causing adverse effects in normal tissues [139]. Comparative studies on normal cell lines, such as fibroblasts or epithelial cells, suggest that both polymers exhibit good biocompatibility at optimized concentrations, but their impact varies depending on factors such as polymer composition, degradation rate, and formulation characteristics. Understanding these differences is essential for designing safer and more effective drug delivery systems.

### 4.2. Glutathione (Redox-Responsive) CS-Based Delivery Systems

Compared to the normal physiological environment, tumors exhibit certain particularities that need to be considered when new drug delivery systems are synthesized. As mentioned before, those particularities refer to low pH, elevated ROS levels, hypoxia, and specific enzyme systems, providing diverse stimuli for the design of polymeric matrices. The possibility of developing GSH-sensitive transport systems was explored by Li et al. in their study [140]. They synthesized a nanosystem based on carboxymethyl-CS in which Rose Bengal (RB) and oxymatrine (OMT) were incorporated to achieve a synergistic anticancer action. RB co-encapsulation has also been evaluated in previous research and proved to be effective in combination with doxorubicin in photodynamic therapy [141]. The same principle was applied in the aforementioned study for the treatment of oral cancer. That NPs system was able to generate a large amount of singlet oxygen, exhibiting a photodynamic effect. It was also able to inhibit the PI3K/AKT signaling pathway in oxidative stress and induce tumor apoptosis through mitochondria-related pathways.

In addition to targeting tumors, current research in cancer therapies also focuses on overcoming MDR and preventing relapses. An innovative CS-based formulation of that type has been proposed by Zhao et al. as a GSH/pH-responsive dual CS-based matrix for reprogramming M2 macrophages and overcoming cancer chemoresistance [142]. The proposed matrix has been designed for the co-delivery of DOX and resiquimod (R848) as a GSH and pH dual-responsive nanoplatform (GPNP) against treatment-resistant cancer cells. Considering the core–shell structure of GPNP, it will detach its polymer shell in a sialic acid-rich medium with the release of R848 in the microenvironment of the tumor. There, M2 macrophages will be reprogrammed into M1 macrophages, exposing the core CS(DOX)-PBA (Phenylboronic Acid) to kill MCF-7/ADR cells. Drug efflux would be diminished by suppressing p-glycoprotein, thus further contributing to overcoming MDR. In addition, the efficacy of the proposed system is also supported by in vitro studies on MCF-7/ADR cells on which apoptosis was induced by reprogramming M2 macrophages.

A recent study led by Zeng and coworkers aimed to synthesize GSH-responsive polymeric micelles that improved the therapeutic response to DOX and reduced metastasis in breast cancer [143]. Considering the hypoxic nature of the TME, it is necessary to introduce a nitric oxide (NO) donor into the structure that dilates blood vessels, mitigates hypoxia, and regulates the TME, thereby potentially improving therapeutic efficacy [144,145]. The matrix under consideration was synthesized by linking CS and octadecylamine (ODA) through a disulfide bond, utilizing S-nitroso-N-acetylpenicillamine (SNAP) as a NO donor. The micelles’ response to GSH concentrations varied. Specifically, the disulfide bonds were not sensitive to GSH from the endothelial cells resulting in minimal DOX release. In contrast, tumor cells facilitated DOX release, with a high targeting capacity that alleviated tumor hypoxia, decreased the infiltration of M2 macrophages in tumors, increased the infiltration of M1 macrophages, and remodeled the TME. Therefore, it may serve as a new drug delivery system that can inhibit metastasis.

### 4.3. pH-Modulated Drug Release from CS-Based Delivery Systems

As previously mentioned in the case of inclusion complexes with CDs, the principle of using pH-responsive matrices is based on the replication of pH values between 6.4 and 7.4 to mimic the tumor microenvironment conditions. Most human tumors have pH values in a range from 6.15 to 7.4, while healthy tissues typically have pH values between 7 and 7.4; nevertheless, certain tumors, such as astrocytomas and squamous cell carcinomas, have pH levels below 6 [49,146,147]. Thus, controlled drug release can be achieved by using that strategy in the formulation of new NPs or hydrogels [148].

The various forms of CS that may be used as an effective drug delivery system employ several hydrophilic polymers, such as polyvinyl alcohol, polyvinyl pyrrolidone, or gelatin, as crosslinkers to create a hydrophilic membrane. Those polymers have a pH-controlled swelling degree, allowing them to swell and de-swell in a wide range of pH.

Glutaraldehyde serves as an example of a crosslinker which provides a wide pH range of value for swelling. For example, it was used by Zhao and colleagues to develop CS-based microspheres for the treatment of osteosarcoma encapsulated with 5-FU, paclitaxel (PTX), and Cis-dichlorodiammine-platinum (CDDP) [149]. In vitro studies conducted on the HOS and MG-63 osteosarcoma cell lines showed significant inhibition growth. The designed system had an average size of 532 μm with a good encapsulation efficiency for each drug (72.4 ± 4.3% for 5-FU, 64.6 ± 3.9% for PTX, and 54.8 ± 5.9% for CDDP, respectively). It showed a favorable release profile and a sustained release of the active drugs, particularly in the presence of glutaraldehyde, for which the degradation period during lysozyme degradation extended for eight weeks. The embedded drugs were released as a result of the swelling process due to the acidic pH of osteosarcoma, proving great therapeutic potential for its treatment. Additionally, also used for the treatment of osteosarcoma was a nanosystem proposed by Amiryaghoubi and coworkers containing CS–folate hybrid magnetic NPs loaded with DOX [150]. DOX release profile indicated a significantly elevated release in acidic conditions. The examined nanocarriers were characterized through FT-IR, DLS, XRD, VSM, TEM, and UV-Vis spectroscopy in order to confirm their structures. The cytotoxicity was assessed by the MTT method for DOX alone and for loaded and unloaded NPs. No significant toxicity was observed on the MG-63 and A549 cell lines (MG-63 and A549) for the unloaded polymers, confirming their biocompatibility. The reduced toxicity of the loaded particles, when compared to free DOX, effectively illustrated the controlled release of DOX. Also, important to mention is that DOX-loaded NPs exhibited higher toxicity on MG-63 cells due to their overexpression of folate receptors compared to folate-negative A549 cells.

The same targeting strategy was employed by Shakeran and coworkers while developing CS-modified mesoporous silica nanoparticles (MSNs) for the delivery of MTX intended for breast cancer therapy [151]. The MSNs were modified with 3-triethoxysilylpropylamine (APTES) and then with CS through covalent linkage mediated by glutaraldehyde as a crosslinker. The obtained pH-sensitive polymer with a particle size of 73.2 ± 4.9 nm exhibited encapsulation efficiency of approximately 11–12% for MTX. The release profile from MSN-APTES/MTX showed a burst release, whereas the nanoparticles MSN-APTES/MTX modified with CS demonstrated a reduced burst release. It was demonstrated that below the isoelectric point of CS, the polymer transformed into its hydrophilic protonated form, allowing it to swell and release MTX, achieving higher MTX levels at pH = 6 than at neutral pH, thereby demonstrating the important role that CS had on the release profile of that matrix. Its in vitro cytotoxicity was evaluated via MTT assay on the MCF7 breast cancer cell line and it showed a significant decrease in their viability at a low dose of loaded MTX, making it a promising technology for breast cancer treatment.

Another interesting delivery system has been developed by Zhang et al. [152]. They have synthesized nano-prodrugs to implement the synergistic antitumoral activity of cisplatin (Pt(IV)-1) and demethylcantharidin. Those polymers were obtained by fluorination, as those molecules have unique properties due to C-F bonds, which positively influence their stability, pharmacokinetic profile, and drug permeability [125]. The fluorinated carboxymethyl CS-based nano-prodrugs were prepared via crosslinking between carboxymethyl CS and Pt(IV)-1, stabilized with glutaraldehyde, and via fluorination using heptafluorobutyric anhydride in an aqueous solution. The researchers reported that the nanosystem was a pH/GSH-responsive dual-crosslinked polymer with lower toxicity and immunogenicity with an enhanced cellular uptake for the active drugs, demonstrated via the quantitative analysis of Pt within the tumor cells (HepG2, MCF-7, A549, and H22).

Table 2 synthetizes the abovementioned findings along with additional studies to provide a clearer perspective on the current research on CS-based delivery systems for various types of cancer. Additional in vivo studies are required to confirm the potential of the proposed nanosystems.

## 5. Summative Discussion

CS-based nanoparticulate formulations present a number of advantages that may counteract the limitations of certain active substances, such as low stability and bioavailability. These attractive features are due to its preferential protonation and solubilization in acidic environments, being able to bind to the negatively charged mucus membrane, increasing the retention time and enhancing the probability of their cellular uptake. Due to its unique features, its use as a coating agent enhances the permeation of antitumoral drugs by transiently opening the tight junction between epithelial cells [157].

CDs, whether naturally occurring or chemically modified, are used to improve the bioavailability of antitumor drugs by forming inclusion complexes that increase water solubility. Conversely, CD-based polymers have the capability to facilitate the membrane transport and stability of encapsulated substances. Consequently, the pharmacokinetic profile of the drug will be changed, the encapsulation will occur in greater proportion and its release will be controlled leading to improved efficacy. Nanoparticulate forms (such as nanosponges, polymeric nanoparticles, graphene nanoparticles, and silica nanoparticles) enhance the stability of many pharmaceutical ingredients, offering the benefits of biodegradability and decreased toxicity. Another advantage of using CDs as components of nanocarriers is to facilitate selective tumor uptake, attributable to the increased glucose consumption by the neoplastic cells [158].

CSNPs and CDNPs control drug delivery by means of their pH sensitivity, redox responsiveness, folate sensitivity, surface flexibility, and mucoadhesive and penetration properties. CS derivatives further enhance the efficiency of anticancer drugs by increasing their retention time, accumulation, cellular uptake, and cytotoxicity to tumor cells as shown in the abovementioned studies.

It is crucial to understand which components of the tumor microenvironment are essential in cancer cell growth and the formation of barriers that impede the efficacy of nanomedicines. The GSH level is considered one of the critical factors, contributing to resistance to many therapeutic approaches including ferroptosis therapy, platinum-based chemotherapy, photodynamic therapy, and radiotherapy. The depletion of GSH could overcome the resistance of solid tumors and facilitate the delivery of therapeutic agents to enhance ferroptosis and immunotherapy across distinct cancer types [159]. This mechanism finds its application in in vitro and in vivo hypoxic cancers with outstanding specificity and biological safety as the authors of [160] demonstrated in their study. Elevated GSH levels are linked with tumor progression and increased multidrug resistance. The role of GSH in the development of drug resistance across various cancers (such as melanoma, hepatocarcinoma, bone marrow, breast, colon, pancreatic, and lung cancers) is established by its antioxidant properties. Thus, controlling glutathione synthesis by inhibiting key enzymes and/or precursors may be a promising strategy to counteract cancer progression [161,162].

Certainly, there are supplementary stimuli with research potential that can facilitate the drug’s release following activation, including reactive oxygen species (ROS), various enzymes, ATP level, and hypoxia. To emphasize the significance of utilizing these stimuli, we present several significant aspects that have not been comprehensively analyzed yet are particularly crucial for future research.

Among them, significant potential exhibits the presence of ROS for which ROS-responsive drug delivery systems have been developed due to ROS hyperproduction at the tumor site. Promising outcomes have been observed in thioether-based materials, diselenide linkages, thioketal-based materials, boronic ester-based, and polyoxalate- based materials [163]. Enzymes have a crucial role in every stage of cancer growth, from angiogenesis to metastasis. Considering this, many tumor hyperexpressed enzymes but with low concentrations in healthy tissues, such as matrix metalloproteinases and cathepsins, offer the possibility of enzyme-responsive drug delivery systems [164].

Another stimulus of interest could be the elevated levels of ATP in tumor cells, which arise from the need for rapid tumor proliferation, potentially serving as a valuable trigger for responsive nanomedicines. Although ATP is the primary energy source for all living cells in the body, the relatively higher level than that in other normal tissues complicates the formulation of drug delivery systems due to the potential for non-specific drug release [165].

Finally, the exploitation of hypoxia is also a viable option to consider in the development of future drug delivery systems utilizing azo linkers, nitrobenzyl alcohol, and nitroimidazoles [166]. The main challenge lies in the different levels within the same tumor tissue, complicating the preferred delivery of the drug to the central region, where hypoxia is more pronounced, as opposed to the peripheral area.

Nevertheless, these matrices exhibit certain limitations, as the toxicity and immunogenicity of the CS nanocarriers towards more normal cell lines need to be studied more, particularly to investigate their long-term impact, which will probably be elucidated in the forthcoming years of research.

Multiple constraints must be addressed before GSH-mediated nanomedicines can receive clinical approval. Several cancer cell types do not induce apoptotic responses following the chemical depletion of GSH. Considering the variability of GSH levels among various tumor types and their development stages is critical, as it can significantly affect the efficiency of GSH-responsive drug delivery systems. Furthermore, GSH depletion would lead to an upregulation of GSH-related synthase and dynamically restore the local GSH level. It is not enough to consider only its antioxidant capacity, but a broader vision is needed that includes other metabolic functions that are not yet completely understood in order to synthesize multifunctional nanoplatforms that would not interfere with normal functions and could integrate both diagnostic imaging and therapy.

It is also important to mention that our understanding regarding the function of GSH, FA, and pH responsiveness correlated with tumor progression is predominantly derived from data obtained from tumor or stromal cells cultured in isolation under in vitro conditions, requiring caution when extrapolating these findings to the intricate in vivo tumor microenvironment. Future research focuses on creating tumor organoid models that could replicate more accurately redox processes that occur in the tumor system, and continued research in this domain is necessary.

Another challenge in formulation that must be addressed is the poor penetration and distribution of nanocarriers in heterogeneous tumor tissue. This arises from various factors, including limited blood supply, high-density tumor cells and extracellular matrix, and high interstitial fluid pressure. Traditional strategies such as the modulation of tumor microenvironments and optimization of nanoparticle properties have been reported to improve tumor penetration of nanomedicines; nevertheless, these conventional approaches still have their limitations. Developing nanocarriers is a highly intricate process, as their activity and transformation within the body are not fully comprehended. Research remains necessary to improve the depth of penetration which is currently modest because of the complexity of biological barriers. For example, modulating the physical properties of nanoparticles compromises multiple factors, as the optimized properties influence distinct delivery cascades in conflicting manners. One such example would refer to small nanoparticles that would exhibit a better penetration efficacy; yet following in vivo administration, they would rapidly be removed by renal clearance.

Targeting ligands such as antibodies, peptides, or aptamers are often conjugated to chitosan- or cyclodextrin-based carriers to enhance selective delivery to cancer cells expressing specific surface receptors. While this strategy can improve therapeutic efficacy and reduce systemic toxicity, it also introduces complexity in synthesis and characterization. Over-functionalization may affect the particle’s size, charge, and circulation time. Furthermore, certain ligands may trigger immune recognition or rapid clearance. Thus, achieving the right balance in ligand density and orientation is essential to preserve both stability and targeting capability.

Recent strategies with broad applicability across multiple types of nanoparticles and tumors are being implemented to address these issues. They include enhancing transcellular transport by the conjugation or co-administration of tumor-specific penetration peptides or by using stimuli-responsive nanoplatforms, as well as developing transformable nanoparticles via PEGylation. Another approach is to avoid relying on tumor penetration and identify alternative methods to bypass these barriers. Recent research has focused on nanoparticle-based immunotherapy, which involves the activation of the immune system by the co-delivery of immune agonists, tumor-specific antigens, and photothermal agents, resulting in tumor regression without necessitating deep tumor penetration [167].

Despite their promising properties, the clinical translation of CD-/CS-based delivery systems is still hindered by several practical challenges. These include difficulties in scaling up production while maintaining batch-to-batch consistency and concerns related to the long-term physical and chemical stability of formulations. Regulatory approval requires comprehensive preclinical data on pharmacokinetics, safety, and toxicology, which are often lacking. Additionally, tumor heterogeneity and individual differences in microenvironmental conditions can affect therapeutic efficacy. Modifications like ligand functionalization, though useful for targeting, may raise immunogenicity risks. Therefore, translational success will depend on designing standardized manufacturing protocols, performing thorough biological assessments, and validating performance in clinically relevant animal models.

Looking ahead, future research should aim to make CD-/CS-based drug delivery systems more clinically relevant by incorporating advanced functionalities. Promising directions include dual-ligand modification, tumor-penetrating peptides, and environmentally responsive features that improve selective targeting and therapeutic efficiency. Additionally, theranostic approaches that pair drug delivery with imaging functions may offer real-time monitoring and better treatment personalization. To ensure these systems are translatable, comprehensive in vivo testing—focusing on pharmacokinetics, biodistribution, and long-term safety—remains a priority. These efforts will be essential for transitioning these innovative systems from the lab bench to the clinical bedside.

## 6. Conclusions

The distinctive properties of CDs and CS discussed above are essential for the development of innovative drug delivery that can lead to a significant improvement in therapeutic success in oncology patients. Their application covers a wide range of cancer types, including colorectal, liver, gastric, pancreatic, esophageal, and oral cancers.

The primary benefits of these nanosystems are prolonged circulation times, improved cellular internalization of loaded drugs, selective uptake by the targeted cells, diminished side effects, enhanced apoptosis rates, and increased tumor suppression rates demonstrated by in vitro and in vivo studies, predominantly conducted on tumor-bearing mice.

Nevertheless, as indicated in the aforementioned studies, these systems have not yet been included in human therapies as they are not FDA-approved. Their efficacy is, however, confirmed on cancer cell lines and animals in preclinical trials, and from the multitude of nanoparticulate therapeutic systems, the most promising ones need to advance into phase I, II, and III clinical trials for a faster introduction into therapeutic regimens. In conclusion, by highlighting the most recent findings involving the synthesis of stimuli-responsive nanosystems with CS and CDs, the present work served as an overview for future research to broaden the therapeutic possibilities of cancer.

New drug design technologies based on machine learning will particularly improve drug targeting techniques and will lead to innovative, much more precise, and sensitive treatments, as well as facilitate the development of safe multimodal cancer therapies.

Glutathione, folic acid, pH, and other important stimuli warrant a significant role in these advancements. Disturbances in either glutathione, folic acid, or pH homeostasis have been well documented in cancer cells; thus, these three factors have been often targeted when developing new delivery systems for cancer drugs. Such carrier systems achieve higher intracellular penetration of the administered drugs, rapid and actively targeted drug release, and drug accumulation in tumor cells to enhance therapeutic effects and decrease drug resistance. Also, they determine reduced systemic drug distribution with decreased toxicity for healthy cells and increased compliance to treatment.

In conclusion, by highlighting the most recent studies involving the synthesis of stimuli-responsive CS- and CD-based nanosystems that encapsulate antitumor drugs, the present work has provided an overview to stimulate further research aimed at increasing the therapeutic efficacy of antitumor drugs by providing targeted and personalized treatment.

## Figures and Tables

**Figure 1 pharmaceuticals-18-00564-f001:**
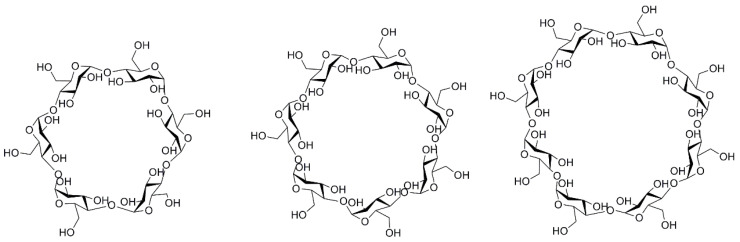
Structures of α-CD, β-CD, and γ-CD.

**Figure 2 pharmaceuticals-18-00564-f002:**
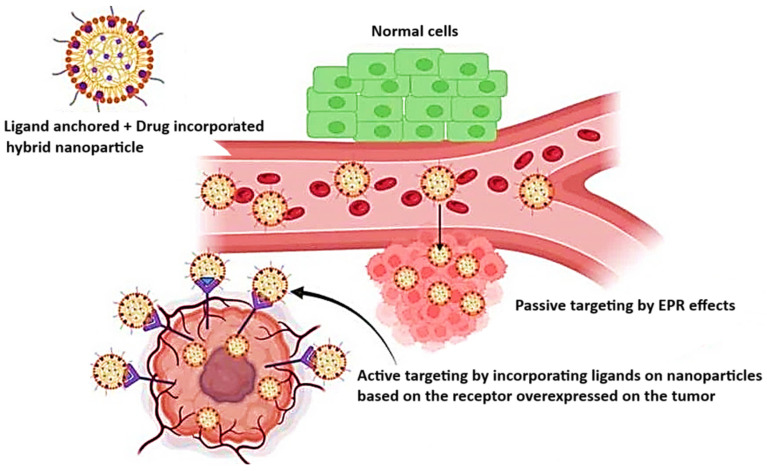
Schematic representation of the passive targeting (EPR effect) and active targeting of the tumor via multifunctional NPs. Reprinted from an open access source [47].

**Figure 3 pharmaceuticals-18-00564-f003:**
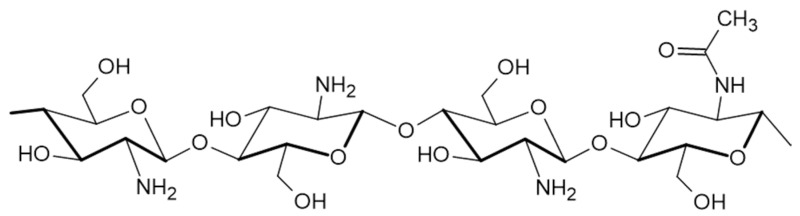
Structure of chitosan.

**Figure 4 pharmaceuticals-18-00564-f004:**
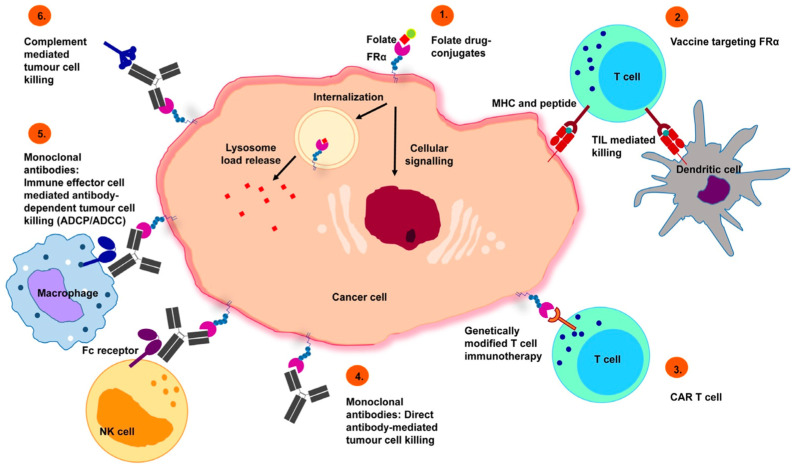
Prospective therapeutic approaches targeting FRα: 1. Folate drug-conjugates—conjugated chemotherapeutic agents designed for FRα targeting. 2. Vaccines targeting FRα—autologous dendritic cells engineered with FRα mRNA initiate an anti-FRα immune response, mediated by T-cells. 3. Chimeric antigen receptor (CAR) T cells—CAR T cells recognize FRα and trigger tumor cell killing. 4. Monoclonal antibodies—specific recognition of FRα inhibits the downstream signaling events that cause tumor cell death. 5. Monoclonal antibodies (immune effector cell engagement)—antibodies link FRα-expressing tumor cells with immune effector cells that bear Fc receptors, potentiate effector cell activation, and target-neutralizing functions. 6. Complement system activated by chimeric anti-folate receptor antibodies designed as an efficient effector system for carcinoma control. Reprinted from an open access source [72].

**Figure 5 pharmaceuticals-18-00564-f005:**
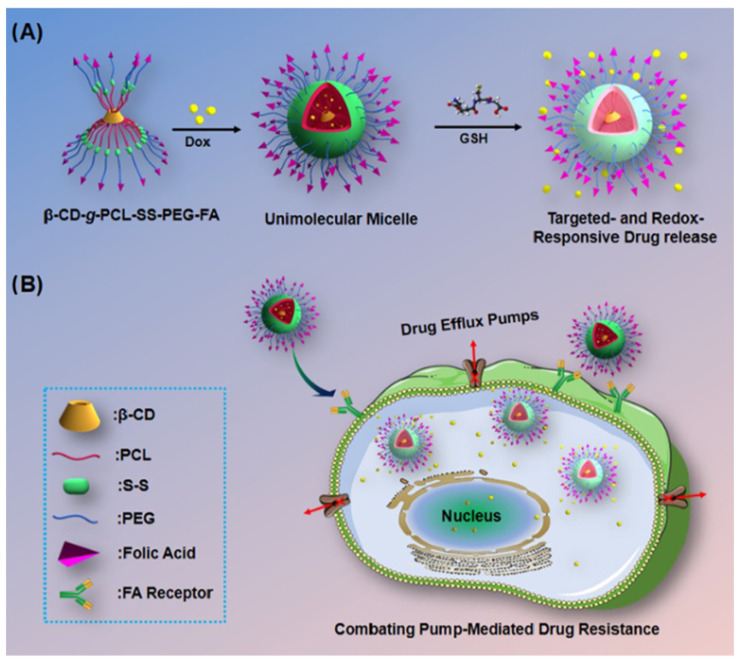
(**A**) Schematic illustration for the design of targeting and GSH-responsive unimolecular micelles. (**B**) Drug-loaded unimolecular micelles show targeted delivery and enhanced therapeutic efficiency against drug-resistant cancer cells. Reprinted from an open access source [83].

**Figure 6 pharmaceuticals-18-00564-f006:**
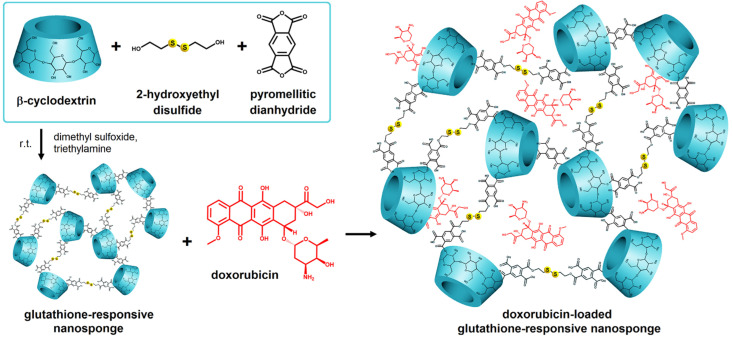
Scheme of the doxorubicin-loaded glutathione-responsive nanosponge (Dox-GSH-NSs) synthesis by Daga et al. Reprinted from an open access source [84].

**Figure 7 pharmaceuticals-18-00564-f007:**
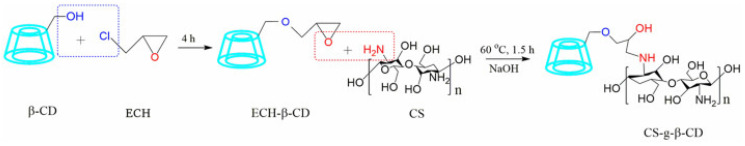
The two successive reactions via the ECH-assisted grafting of CS to β-CD are achieved. Reprinted from an open access source [109].

**Figure 8 pharmaceuticals-18-00564-f008:**
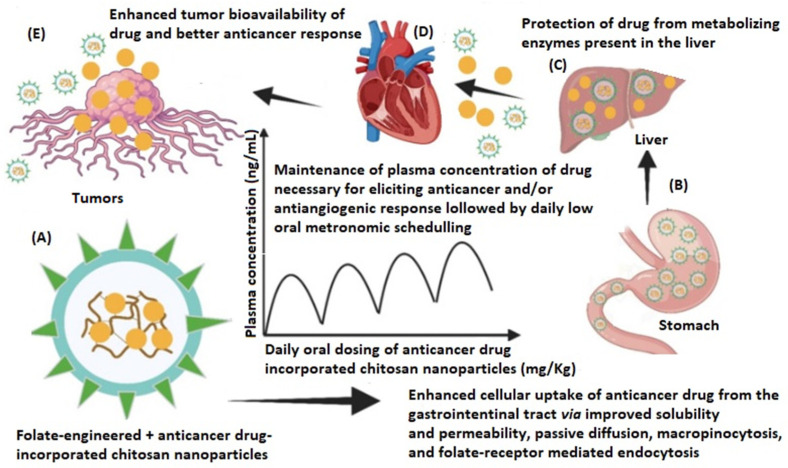
Schematic representation of the application of folate-engineered NPs loaded with an antitumoral drug: (**A**) design of folate-engineered CSNPs; (**B**) enhanced permeation from the stomach due to the improved solubility and permeability of drug after being incorporated into the CSNPs delivery system; (**C**) the drug-loaded NP metabolism is prevented by the liver enzymes, thereby resulting in lower dosing frequency and diminished toxic effect; (**D**) biodistribution of drug-loaded NPs; (**E**) the drug-incorporated CSNPs permeate the tumor via folate receptor interaction, passive diffusion, and EPR effect. Reprinted from an open access source [47].

**Figure 9 pharmaceuticals-18-00564-f009:**
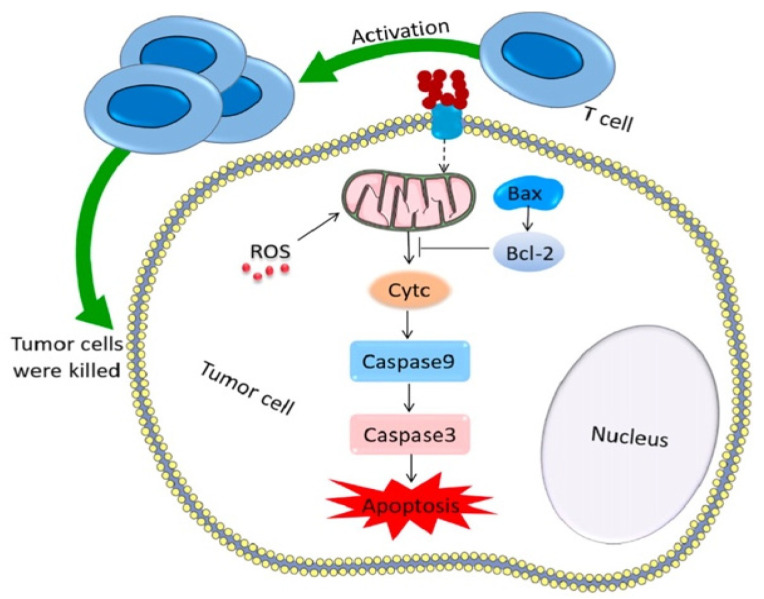
Antitumor mechanism for inducing tumor cell apoptosis. Reprinted from an open access source [132].

**Table 1 pharmaceuticals-18-00564-t001:** The application of CDs for delivery of therapeutic compounds in cancer therapy.

Drug Delivery System	Carried Agent(s)	Type of Cancer	Targeting Receptor/ Stimuli	Main Observations	Ref.
Folate-conjugated β-CD-polycaprolactone	curcumin	cervical cancer	pH temperature folate receptor	Self-assembly curcumin-loaded nanoparticles (FA-Cur-NPs). Significantly higher release profile at pH = 6.4 (PBS) than at pH = 7.4. Significant accumulation of FA-Cur-NPs in the tumor site and excellent antitumor activity on the HeLa xenograft mice model with marked tumor growth inhibition. Hematoxylin-eosin (H&E) staining of the major organs showed no discernible injuries after treatment with FA-Cur-NPs.	[78]
PEG-coated β-CD-based NPs (CD-PEG-FA.Rg3.QTN)	ginsenoside (Rg3) quercetin (QTN)	colorectal cancer	folate receptor	Encapsulation efficiency (EE%) of more than 95%. Synergistic anticancer effect of the two encapsulated substances. Alterations to the immunosuppressive character of TME when combined with programmed death-ligand 1 (PD-L1) therapy. H&E staining results show that no significant histological changes were found in major organs of mice following i.v. injection of targeted co-formulation as compared to PBS. No significant hematological toxicity and liver/kidney injuries were caused by targeted co-formulation as compared to PBS, indicating that no systemic toxicity was caused by targeted co-formulation under the tested doses.	[79]
Polymeric micelle, β-CD-g-PCL-SS-PEG-FA	doxorubicin (DOX)	cervical cancer/MDR1 hepatocellular carcinoma/MDR1	GSH folate receptor	Reuptake of FA-containing compounds induced by the overexpressed folic acid receptors. Released DOX from the micelles due to the cleavage of disulfide bonds induced by high glutathione (GSH) concentration. Potential for drug-resistant cancers. In FR highly expressed HeLa/MDR1 cells, micelles with FA group linkage were optimal for drug uptake.	[83]
β-CD-based nanosponge (NS)	doxorubicin (DOX) 6-coumarin	hepatocellular carcinoma	GSH pH	Intracellular accumulation of NS, demonstrated by rapid fluorescence (15 min). NS bypasses the P-gp efflux pump. Lack of toxicity of unloaded GSH-NS with the potential of incorporation of other drugs. Inhibitory capacity on HepG2 cell line. Cytotoxic action of unloaded GSH-NS higher on the DU145 prostate cancer cell line than on HCT116 colorectal cancer cells due to its higher GSH content.	[85,87]
ROS-generating NPs (TA-β-CD@DHA NPs)	dihydroartemisinin (DHA)	colon carcinoma	pH GSH	Cell death caused by ROS and lipid peroxidation. In vitro cytotoxicity of DHA established on CT26 cell line. In vivo antitumor activity established on CT26 tumor-bearing mice. No visible pathological inflammation or tissue damage in the H&E stained sections of main organs was observed in all the treatment groups, confirming great compatibility within therapeutic doses.	[91]
Pt(IV)-SSNPs	cisplatin as Pt(IV)-ADA2 prodrug	colon carcinoma	GSH	Adamanthyl-platinum IV prodrug, Pt(IV)-ADA2, is reduced to the active cisplatin form in a strongly reducing environment, such as that of tumor cells. Tumor inhibition confirmed by MTT assay on CT26 cell line and in vivo studies on tumor-bearing mice. Pt(IV)-SSNPs exhibited the strongest suppression effect on tumor progression without causing obvious side effects in comparison with cisplatin, which inhibited tumor growth effectively, but it caused severe side effects, such as weight loss, nephrotoxicity, and even death.	[93]
5-FU/IL-2/CD nanoplexes	5-fluorouracil (5-FU) interleukin-2 (IL-2)	colorectal cancer	pH	Almost complete EE% for IL-2 (99.8%). Cell viability assessed by LDH method. Antitumor activity of CD nanoplexes confirmed in CT26 tumor model mice. In vivo studies were conducted on healthy mice to monitor ALT (alanine aminotransferase) and AST (aspartate aminotransferase), the primary enzymes used to measure liver injury and toxicity; CD2 derivative was selected, as CD1 and CD3 derivatives might induce liver damage and were classified as toxic. Histopathological evaluation of in vivo cancer nodules showed fewer foci developed in mice treated with 5FU/IL2/CD2 nanoplex compared with mice treated with only 5-FU solution.	[98]
Epichlorohydrin (EPI)- β-CD hydrogel	5-fluorouracil (5-FU) methotrexate (MTX)	breast cancer	pH	Cytotoxicity investigated in MC-7 breast tumor cell line. Histopathologic analysis showed the improvement of cancer manifestation (swelling and inflammation) after intratumor injection of loaded hydrogel systems.	[110]
Sulfobutyl ether-β-cyclodextrin complex (MF-SEBCD)	5-fluorouracil (5-FU) methotrexate (MTX)	colorectal cancer	pH	A 92-fold enhancement in the solubility of MF (5-fluorouracil-methotrexate conjugate). Two-fold extension of the half-life of 5-FU. Antitumor activity confirmed on MC38 mouse ectopic colon cancer model. In the high-dose physical mixture group, levels of ALT, AST, and GGT were higher compared with the saline group, indicating some hepatic toxicity, while the mice treated with MF-SEBCD did not exhibit any significant differences in complete blood count and blood biochemical parameters, indicating that it does not cause systemic toxicity.	[119]
pCyD/SFN and oCyD/SFN	sorafenib (SFN)	breast cancer hepatocellular carcinoma gastric cancer melanoma colon carcinoma	pH	Antiproliferative effects demonstrated on the following cell lines: MCF7, HGC-27, HepG2, SKMel-28, and K1. The inclusion complex triggered apoptosis with a lower in vivo toxicity compared to the free drug. Histopathological analysis showed the following results: free SFN administration led to extensively altered parenchymal areas of the lung with compromised alveolar morphology, the collapse of the alveolar component, and erythrocyte congestion in comparison with oCyD/SFN, which caused only occasional alterations of parenchyma, accompanied by reduced erythrocyte congestion. The liver sections had a regular appearance after oCyD/SFN and pCyD/SFN administration, while after the administration of SFN (per os), dilated sinusoidal areas, vessels frequently congested by erythrocyte and occasionally amorphous material were observed. In kidney sections, a greater presence of proximal tubules with altered epithelium and vessels congested by erythrocytes was observed in mice treated with oCyD/SFN and pCyD/SFN.	[120]
β-cyclodextrin or hydroxypropyl-β-cyclodextrin thermosensitive gel	5-fluorouracil (5-FU)	HPV-induced cervical cancers	pH temperature	Complete release of 5-FU from gels was obtained with both β-CD and HP-β-CD. Cytotoxicity studies against HeLa human cervical carcinoma cells demonstrated that 1% 5-FU:CD complexes were equally effective as 1% free 5-FU.	[122]

QTN: quercetin; DOX: doxorubicin; DHA: dihydroartemisinin; 5-FU: 5-fluorouracil; IL-2: interleukin-2; MTX: methotrexate; SFN: sorafenib.

**Table 2 pharmaceuticals-18-00564-t002:** Reported research on the application of various CS-based delivery systems to treat different types of cancer.

Drug Delivery System	Carried Agent(s)	Type of Cancer	Targeting Receptor/Stimuli	Main Observations	Ref.
Chitosan (CS)-poly lactic-co-glycolic acid (PLGA)-folic acid (FA) nanocarrier (CPSF)	sorafenib (SFN)	lung cancer	pH folate receptor	The MTT assay demonstrated a cell viability of 13% after 24 h treatment with 400 nM CPSF in A549 cancer cells while it was 78% in MSC normal cells. The qRT-PCR revealed >8-fold and 11-fold increase for Caspase9 and P53 apoptotic genes after 5 h treatment with 100 nM (IC50) CPSF.	[70]
Poly(lactic-co-glycolic acid), chitosan, polyethylene glycol, and folic acid microbubbles	methotrexate (MTX)	breast cancer	folate receptor	Higher internalization of MTX due to the receptor-mediated opsonization. The active targeting ability of the FA-nanodroplets with ultrasound radiation was confirmed by flow cytometry and the apoptosis properties of MTX-loaded nanodroplets were investigated via DAPI staining assay and MTT assay on the MCF-7 breast cancer cell line.	[129]
Folate-conjugated chitosan nanoparticles (FCCNPs)	cytarabine	breast cancer	folate receptor	NPs prepared by ionic gelation using tripolyphosphate (TPP) as a crosslinker. The expression of apoptotic genes (*bax*, *cyt c*, and *cas 9*) increased after 72 h, inducing apoptosis. Cytotoxicity assessed on the MCF-7 cell line with favorable results. In vivo evaluation of FCCNP in a suitable animal model is essential to confirm its improved anticancer effects and safety.	[131]
FA-CS-modified PLGA nanoparticles (C-FA-PNPs)	irinotecan (IRI) and quercetin (QTN)	colon cancer	folate receptor	Controlled release from the C-FA-PNP matrix for the two active substances. Cytotoxicity studies conducted on the Caco-2 cell line have demonstrated lower viability for the nanoparticulate formulation. Efficacy of the formulation was assessed in colon cancer-induced rats. Histopathological deviations were assessed by using H&E staining and they showed severe detrimental effects on major organs: brain—degenerated neurons, vacuolation, necrosis, liver—dilated blood vessels and sinusoids, kidney—degenerated tubules, stomach—damaged glands with dilated lumen in mucosa, intestinal tissues—chronic inflammatory cells. These detrimental effects were comparably lesser in the tissues obtained from animals treated with both the formulation groups in comparison with the marketed product.	[133]
Carboxymethyl-CS-nanosystem	Rose Bengal (RB) and oxymatrine (OMT)	oral cancer	GSH	The OMT/CMCS-CYS-RB NPs system generates a large amount of singlet oxygen, exhibiting a photodynamic effect. Inhibits the PI3K/AKT signaling pathway in oxidative stress and induces tumor apoptosis.	[140]
CS-based GPNPs (GSH and pH dual-responsive nanoplatform)	doxorubicin (DOX) and resiquimod (R848)	breast cancer, treatment-resistant	pH GSH	Chitosan-based matrix designed for reprogramming M2 macrophages and overcoming cancer chemoresistance. In vitro studies on MCF-7/ADR cells on which apoptosis was induced by reprogramming M2 macrophages.	[142]
CS polymeric micelles (TSCO–SS–ODA/DOX)	doxorubicin (DOX)	breast cancer	GSH	In tumor cells, the cleavage of the disulfide bond led to an efficient release of DOX. Considering the hypoxic nature of the TME, it is necessary to introduce a nitric oxide (NO), in this case, S-nitroso-N-acetylpenicillamine (SNAP) was used.	[143]
CS Microspheres	5-fluorouracil (5-FU), paclitaxel (PTX), and Cis-dichlorodiammine-platinum (CDDP)	osteosarcoma	pH	Sustained release of the active drugs. Significant inhibition growth on the HOS and MG-63 osteosarcoma cell lines. Significantly increased BSX and Caspase-3 apoptotic proteins on both cell lines.	[149]
CS-based magnetic nanoparticles (MNPs) conjugated with FA and functionalized with succinic anhydride (CS-FA/CS-SA@MNPs)	doxorubicin (DOX)	osteosarcoma	pH folate receptor	DOX release profile at various pH levels demonstrated an enhanced release of DOX under acidic conditions. MG-63 cells, which partially express folate receptors (FRs), particularly FR-α, showed higher cellular uptake of the DOX-loaded NPs compared to the FR-negative lung cancer A549 cells. MTT assay demonstrated that DOX-NPs had increased effectiveness and greater cytotoxic effects compared to DOX alone.	[150]
CS-modified mesoporous silica NPs (MSNs) modified with 3-triethoxysilylpropylamine (APTES)	methotrexate (MTX)	breast cancer	pH	MSN-APTES/MTX modified with chitosan demonstrated a reduced burst release. In vitro cytotoxicity was evaluated via an MTT assay on the MCF7 breast cancer cell line and showed a significant decrease in their viability at a low dose of loaded MTX.	[151]
Fluorinated Carboxymethyl chitosan-based nano-prodrugs (F-NGs)	prodrug (Pt(IV)-1)-containing cisplatin (DDP) and demethylcantharidin (DMC) at a molar ratio of 1:2	hepatocellular carcinoma lung cancer breast cancer	pH GSH	Synergistic chemotherapy. pH/GSH-responsive dual-crosslinked polymer with lower toxicity and immunogenicity with an enhanced cellular uptake for the active drugs, demonstrated via quantitative analysis of Pt within the tumor cells (HepG2, MCF-7, A549, and H22). In vivo tumor inhibition detected on H22 tumor-bearing mice. Blood samples indicated an AST level in F-NG groups similar to those in control.	[152]
FA-CSNPs	thymoquinone	ovarian cancer	folate receptor	Radioiodinated folic acid NPs with increased targeting ability on ovarian cancer demonstrated on SKOV-3 and Caco-2 cells at lower IC50 values. Cellular uptake for NPs in SKOV-3 cells was twice as high as in the Caco-2 cell line.	[153]
CS-FA-Hesperetin (CFH) NPs	doublecortin like kinase 1 antibody (DCLK1) and hesperetin	colon cancer	folate receptor	CFH-DCLK1 NPs effectively suppress the expression of cancer stem cell markers such as STAT1, NOTCH1, and DCLK1.	[154]
Lipid CS hybrid NPs	cisplatin	breast cancer ovarian cancer	folate receptor	Higher cell uptake for FLPHNPs due to enhanced uptake by folate receptor-mediated endocytosis. FLPHNPs showed a 2.4-fold higher uptake in comparison to the LPHNPs.	[155]
CS/stearic acid NPs (CSSA NPs)	doxorubicin (DOX) curcumin	colorectal cancer	GSH	High encapsulation efficiency for both active substances (86.6% for DOX and 82% for curcumin, respectively). Almost complete drug release was achieved in the presence of 10 mM GSH. Higher cancer cell killing efficiency in comparison to each free drug for CSSA NPs, confirmed on HCT116 cell line.	[156]

CS: chitosan; 5-FU: 5-fluorouracil; PTX: paclitaxel; CDDP: Cis-dichlorodiammine-platinum; NPs: nanoparticles; GPNPs: glutathione and pH dual-responsive nanoparticles; DOX: doxorubicin; MTX: methotrexate; sorafenib: SFN; IRI: irinotecan; QTN: quercetin; RB: Rose Bengal; OMT: oxymatrine; R848: resiquimode.

## Data Availability

No new data were created or analyzed in this study. Data sharing is not applicable to this article.

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
