# Peer review of "Smart Drug Delivery Systems Based on Cyclodextrins and Chitosan for Cancer Therapy"

_pharmaceuticals, 2025, doi:10.3390/ph18040564_

Round 1

Reviewer 1 Report

Comments and Suggestions for Authors

The review covers an essential topic regarding Cyclodextrins and Chitosan as drug delivery systems and the novel ways of applications in terms of enhancement of cancer cells targeting following physical-chemical characteristics. There is a need to discuss this important topic and to focus the interest of novel nano-tools that have a highly promising effect compared to other nano carries.

Comments on the Quality of English Language

The text in general is good, but it needs a deep linguistic review from a native speaker reviewer for language optimization in terms of selecting right vocabulary and grammar that can give a reader a better scientific information.

Reviewer 2 Report

Comments and Suggestions for Authors

This review discusses the application of cyclodextrin (CD)- and chitosan (CS)-based smart drug delivery systems (SDDS) in cancer therapy, emphasizing stimuli-responsive platforms (e.g., pH, redox, folate receptors) to enhance tumor-targeted drug release. The authors highlight the potential of these systems to address limitations of conventional therapies, such as poor solubility, bioavailability, and systemic toxicity of antineoplastic drugs. The topic is  aligns with advancements in nanomedicine and precision oncology. However, many revisions are required

  • In the abstract, please nominate time-frame for the period covered by this review
  • In the abstract, add inclusion criteria, and methodological approach (e.g., systematic vs. narrative).
  • In the abstract, provide short sentence about Comparison CD/CS systems with alternative aproaches and explore understudied stimuli.
  • Explain the novelty and originality over old exciting reviews in the literature such as

Najm, A., Niculescu, A.G., Bolocan, A., Rădulescu, M., Grumezescu, A.M., Beuran, M. and Gaspar, B.S., 2023. Chitosan and Cyclodextrins—Versatile Materials Used to Create Drug Delivery Systems for Gastrointestinal Cancers. Pharmaceutics16(1), p.43.

Păduraru, D.N., Niculescu, A.G., Bolocan, A., Andronic, O., Grumezescu, A.M. and Bîrlă, R., 2022. An updated overview of cyclodextrin-based drug delivery systems for cancer therapy. Pharmaceutics14(8), p.1748.

Reference 138 in this review 

  • Line 325, needs appropriate references .
  • Line 347, write full name for PEG
  • Arrange abbreviation list alphabetically
  • Rewrite the following sentence to be more clearer in line 413, which suggested the potential of that system in in therapy for resistant cancers to conventional treatments. . besides, remove extra in
  • The text in section 3 is a little bit long. Reduction is strongly recommended.
  • More illustrative figures are strongly recommended.
  • Future plans should be highlighted more in the discussion
  • In the conclusion , the mention of glutathione, folic acid, and pH (Line 962) needs further explanations. So, it is recommended to briefly explain their roles in tumor microenvironments or targeting.

Reviewer 3 Report

Comments and Suggestions for Authors

1. How do stimuli-responsive drug carrier systems based on cyclodextrins (CDs) and chitosan (CS) achieve controlled and targeted drug release in tumor environments?

2. What is the significance of folate receptor-mediated drug delivery in cyclodextrin-based nanocarriers, and how does it enhance intracellular drug uptake in cancer cells?

3. How does glutathione (GSH) overexpression in cancer cells influence the function of redox-responsive cyclodextrin-based drug delivery systems?

4. How do the molecular weight (MW) and degree of deacetylation (DD) of chitosan affect its drug encapsulation efficiency and controlled drug release kinetics?

5. How do combined delivery strategies, such as the encapsulation of 5-fluorouracil (5-FU) and methotrexate (MTX) in epichlorohydrin-crosslinked β-cyclodextrin hydrogels, enhance therapeutic efficacy and reduce systemic toxicity?

6. What are the main limitations preventing the widespread clinical adoption of cyclodextrin- and chitosan-based smart drug delivery systems in oncology?

7. What are the advantages and challenges of functionalizing cyclodextrin- or chitosan-based nanoparticles with targeting ligands such as aptamers, antibodies, or peptides for active tumor targeting?

Round 2

Reviewer 2 Report

Comments and Suggestions for Authors

The author did all required recommendations. The paper could be accepted in the current form.